# Early detection of cerebrovascular pathology and protective antiviral immunity by MRI

**Li Liu[1]\*, Steve Dodd[1], Ryan D Hunt[1], Nikorn Pothayee[1], Tatjana Atanasijevic[1], Nadia Bouraoud[1], Dragan Maric[2], E Ashley Moseman[3,4], Selamawit Gossa[4], Dorian B McGavern[4], Alan P Koretsky[1]\***

[1]Laboratory of Functional and Molecular Imaging, National Institute of Neurological Disorders and Stroke, National Institutes of Health, Bethesda, United States; [2]Flow and Imaging Cytometry Core Facility, National Institute of Neurological Disorders and Stroke, National Institutes of Health, Bethesda, United States; [3]Department of Immunology, Duke University School of Medicine, Durham, United States; [4]Viral Immunology and Intravital Imaging Section, National Institute of Neurological Disorders and Stroke, National Institutes of Health, Bethesda, United States

**\*For correspondence:**
li.liu3@nih.gov (LL);
koretskya@ninds.nih.gov (APK)

**Competing interest:** The authors declare that no competing interests exist.

**Abstract** Central nervous system (CNS) infections are a major cause of human morbidity and mortality worldwide. Even patients that survive, CNS infections can have lasting neurological dysfunction resulting from immune and pathogen induced pathology. Developing approaches to noninvasively track pathology and immunity in the infected CNS is crucial for patient management and development of new therapeutics. Here, we develop novel MRI-based approaches to monitor virus-specific CD8+ T cells and their relationship to cerebrovascular pathology in the living brain. We studied a relevant murine model in which a neurotropic virus (vesicular stomatitis virus) was introduced intranasally and then entered the brain via olfactory sensory neurons – a route exploited by many pathogens in humans. Using T2\*-weighted high-resolution MRI, we identified small cerebral microbleeds as an early form of pathology associated with viral entry into the brain. Mechanistically, these microbleeds occurred in the absence of peripheral immune cells and were associated with infection of vascular endothelial cells. We monitored the adaptive response to this infection by developing methods to iron label and track individual virus specific CD8+ T cells by MRI. Transferred antiviral T cells were detected in the brain within a day of infection and were able to reduce cerebral microbleeds. These data demonstrate the utility of MRI in detecting the earliest pathological events in the virally infected CNS as well as the therapeutic potential of antiviral T cells in mitigating this pathology.

## Editor's evaluation

This manuscript is of broad interest to researchers studying central nervous system (CNS) infections and associated pathology. Utilizing a rodent intranasal infection model as a route of CNS entry, novel magnetic resonance imaging (MRI) approaches are explored to non-invasively track sites of brain vascular microbleeds and the associated immune cell invasion, particularly virus-specific CD8+ T cells and their relationship to cerebrovascular pathology in living brains. This study is a timely report about how MRI can detect a potential biomarker for early detection of CNS infections and is used as a surrogate readout for treatment efficacy. It will be of interest to research communities involved in immunotherapies across a broad disease continuum as well as imaging physicists.

## Introduction

The CNS is protected by several physical barriers, however, a variety of neurotropic viruses from different families are able to infect the CNS (*Swanson and McGavern, 2015*). Viruses use multiple strategies to access the CNS. Some can infect the brain through the blood brain barrier (BBB) directly, whereas others enter the peripheral nervous system and use axonal transport along motor and olfactory neurons. Infection of hematopoietic cells is also used to enter the CNS (*McGavern and Kang, 2011*).

One common pathology associated with viral infection is bleeding. Two major causes of virus-induced CNS bleeding are direct killing of infected cells and immune-mediated damage (*Ludlow et al., 2016*). Neural and glial inflammatory reactions play an important role in the defense against a viral infection (*Chhatbar et al., 2018*), as do innate and adaptive immune cells recruited from the periphery (*Manglani and McGavern, 2018*). Antiviral T cells recruited into the brain in response to infection can be both helpful and harmful (*Klein and Hunter, 2017*). For example, studies have shown that antiviral memory T cells adoptively transferred into mice persistently infected from birth with lymphocytic choriomeningitis virus (LCMV) can clear virus from the brain without causing immunopathology (*Herz et al., 2015*). However, pathogen-specific CD8 T cells can also trigger fatal vascular breakdown, which has been observed during cerebral malaria (*Belnoue et al., 2002*; *Riggle et al., 2020*; *Swanson et al., 2016*), viral meningitis (*Kim et al., 2009*), and SARS-CoV-2 infection (*Schwabenland et al., 2021*). Thus, immunotherapies designed to treat CNS pathogens face more challenges than those focused on peripheral infections due to the sensitivity of the CNS compartment as well as the unique features and functions of cerebrovasculature.

Magnetic resonance imaging (MRI) has been very useful for the detection of pathological changes in the CNS during viral infection (*Lee et al., 2021*; *Poyiadji et al., 2020*). Microbleeds can be detected sensitively as punctate hypointensity spots on $T2^*$-weighted MRI (gradient-recalled echo or susceptibility-weighted imaging) (*Griffin et al., 2019*). In post-mortem brain tissue from COVID patients, microbleeds were detected by $T2^*$-weighted MRI, which guided the histology studies of cerebrovascular pathology and associated inflammation (*Lee et al., 2021*). Recently, there has been growing interest in using MRI to track cell movements, even at the single cell level (*Pothayee et al., 2017*). Noninvasive MRI based tracking of antiviral T cells in relation to microbleeds should provide additional insights into the mechanisms that result in CNS vascular bleeding during different types of infections.

In this study, we used intranasal infection of mice with vesicular stomatitis virus (VSV) as a model (*Sabin and Olitsky, 1937*) to explore cerebrovascular pathology and antiviral T cells by MRI. The olfactory bulb (OB) serves as the CNS entry point for many neurotropic viruses inhaled into the nose. For example, VSV can infect olfactory sensory neurons that project fibers from the nasal airway to the OB, enabling viral transit within axon fibers and CNS entry (*Chhatbar et al., 2018*; *Moseman et al., 2020*). The main goal of this study was to noninvasively define the temporal and spatial relationship between cerebrovascular breakdown and CD8 T cell infiltration in the virus-infected brain. More specifically, our aims were to: (i) characterize microbleeds as an neuroimaging marker starting at early stages of CNS viral infection by $T2^*$-weighted MRI; (ii) determine whether VSV-induced cerebrovascular damage was caused by the virus or infiltrating immune cells; (iii) evaluate whether adoptively transferred virus-specific CD8 T cells could prevent cerebrovascular damage; (iv) develop an MRI technique to label and track antiviral CD8 T cells; (v) define the relationship between microbleeds and CD8 T cells. Our results demonstrate that VSV promotes extensive bleeding in the OB and other brain regions. Bleeding occurred even in the absence of peripheral immune infiltration, suggesting that the virus alone (or, in combination with brain resident cells) can cause vascular breakdown. Spatially, virus-specific CD8 T cells were observed in brain regions with and without bleeding, and vascular pathology was reduced in mice supplemented with these antiviral cells.

## Results

### Detection of microbleeds in the VSV-infected turbinates and brain by MRI

Upon intranasal VSV inoculation, immune cell accumulation in the turbinates and OB has been reported (*Chhatbar et al., 2018*; *Moseman et al., 2020*). To determine how VSV infection affects

vessel integrity, high resolution T2*-weighted MRI was used to detect bleeding at different time points post-infection (*Figure 1*). Large areas of bleeding were readily detected in the turbinates at day 4 (*Figure 1A*; *Figure 1—figure supplement 1A*). The amount of bleeding, as indicated by hypointense areas (red arrows), increased from day 4–11. Two major sites of bleeding were also detected by MRI in the OB (*Figure 1B–D*; *Figure 1—figure supplement 1B*). Bleeding was observed at the edge of the bulb, in the olfactory nerve layer (ONL) and glomerular layer (GL), and at the center of the bulb near the granule cell layer (GCL). The in-vivo and ex-vivo images on each day were obtained from the same mouse before and after perfusion. In the in-vivo images (upper panel), the asymmetric hypointense region at the edge of the OB indicates the bleeding. As early as day 4, in-vivo MRI revealed microbleeds and vessel rupture starting in the ONL and GL of the OB (red arrows; *Figure 1B and C*; *Figure 1—figure supplement 1B*). Ex-vivo high-resolution MRI detected punctate hypointensities in the ONL and GL on day 4, which confirmed the results of in-vivo MRI. On day 6, punctate hypointensities near the center of the OB were also detected by in-vivo MRI (*Figure 1D*). The number of microbleeds in the OB increased from day 4–11. The increasing amount of hypointensities in the turbinates and OB were quantified as a function of time (*Figure 1E*). A previous study (*Moseman et al., 2020*) reported that VSV infected OB very efficiently and viral titers reached ~$10^4$ plaque-forming units (PFU) on day 1. Viral titer peaked on day 6, ~$10^5$ PFU, before being cleared on day 8. Viral RNA measurements on day 6 and 11 (*Figure 1F*) agreed with this earlier report.

Immunohistochemical (IHC) analyses verified the MRI results by showing that the ONL, GL, and GCL were the major sites of bleeding (*Figure 1G and H*; *Figure 1—figure supplement 1D, E*). These are also the sites where VSV entered the brain and replicated. On day 4, IHC showed microbleeds and VSV in the ONL, GL and external plexiform layer (EPL) of the OB (*Figure 1G*, *Figure 1—figure supplement 1D*). On day 6, microbleeds and VSV were shown at the mitral cell layer (MCL) (*Figure 1 H1*; *Figure 1—figure supplement 1E*) and center of the OB (*Figure 1 H2*). Most of the virus was localized in ONL, GL, and the center of the OB on day 6 when viral titer was the highest (*Figure 1H*). Thus, microbleeds largely co-localized with sites of VSV replication in the brain.

Although VSV replication is usually contained within the OB (*Detje et al., 2009*), we detected vessel breakdown and microbleeds in other parts of the brain by MRI (*Figure 2A and B*; more views are shown in *Figure 2—figure supplement 1A, B*). Punctate hypointensities were detected from frontal brain to midbrain beginning at day 4. In most of the infected mice, microbleeds were detected in the orbital frontal cortex area and somatomotor areas of the cortex around day 4. On day 6, microbleeds were detected in the caudoputamen (CP) and thalamus (TH). From day 8–11, microbleeds were observed from midbrain to hindbrain. IHC verified the presence of microbleeds and VSV in the brain on different days (*Figure 2D*; *Figure 2—figure supplement 1C, D*). The increasing amount of hypointensities in the brain (not including the OB) was quantified as a function of time (*Figure 2C*). For the OB and other brain regions, sites of bleeding were enumerated without accounting for size. This was done to assess the number of new bleeding regions that developed. Foci of bleeding (defined as spots/mm$^3$) were ~100-fold greater in the OB than the brain. For these calculations, the size of mouse OB was assumed to be ~2% of the brain by volume (*McGann, 2017*). The elevated bleeding in the OB is consistent with the fact that VSV is most abundant in this brain region following intranasal inoculation.

## VSV caused vascular endothelium activation and induced cerebral hemorrhage that does not require peripheral immune cells

Significantly increased expression of vascular cell adhesion molecule 1 (VCAM-1) and intercellular adhesion molecule 1 (ICAM-1) were found in the OB on day 6 of infection. VCAM-1 and ICAM-1 vessel surface coverage was around 50% near the ONL/GL/EPL region (*Figure 3A*) and around 30% near the center of the OB (*Figure 3—figure supplement 1A, C*) using anti-CD31 or anti-PECAM-1 to stain the entire vessel. Upregulation of VCAM-1 and ICAM-1 were also found in many other locations in the midbrain and hindbrain on day 6, e.g., thalamus (*Figure 3—figure supplement 1B, C*).

To determine whether the vessel breakdown and microbleeds associated with VSV infection required the peripheral immune system, anti-LFA-1/VLA-4 antibodies were used to block infiltration by peripheral immune cells. LFA-1 and VLA-4 are the ligands for ICAM-1 and VCAM-1, which are expressed on activated T cells, monocytes, and neutrophils (*Yusuf-Makagiansar et al., 2002*). We verified the efficacy of anti-LFA-1/VLA-4 antibodies by cell counting and flow cytometric analysis. The

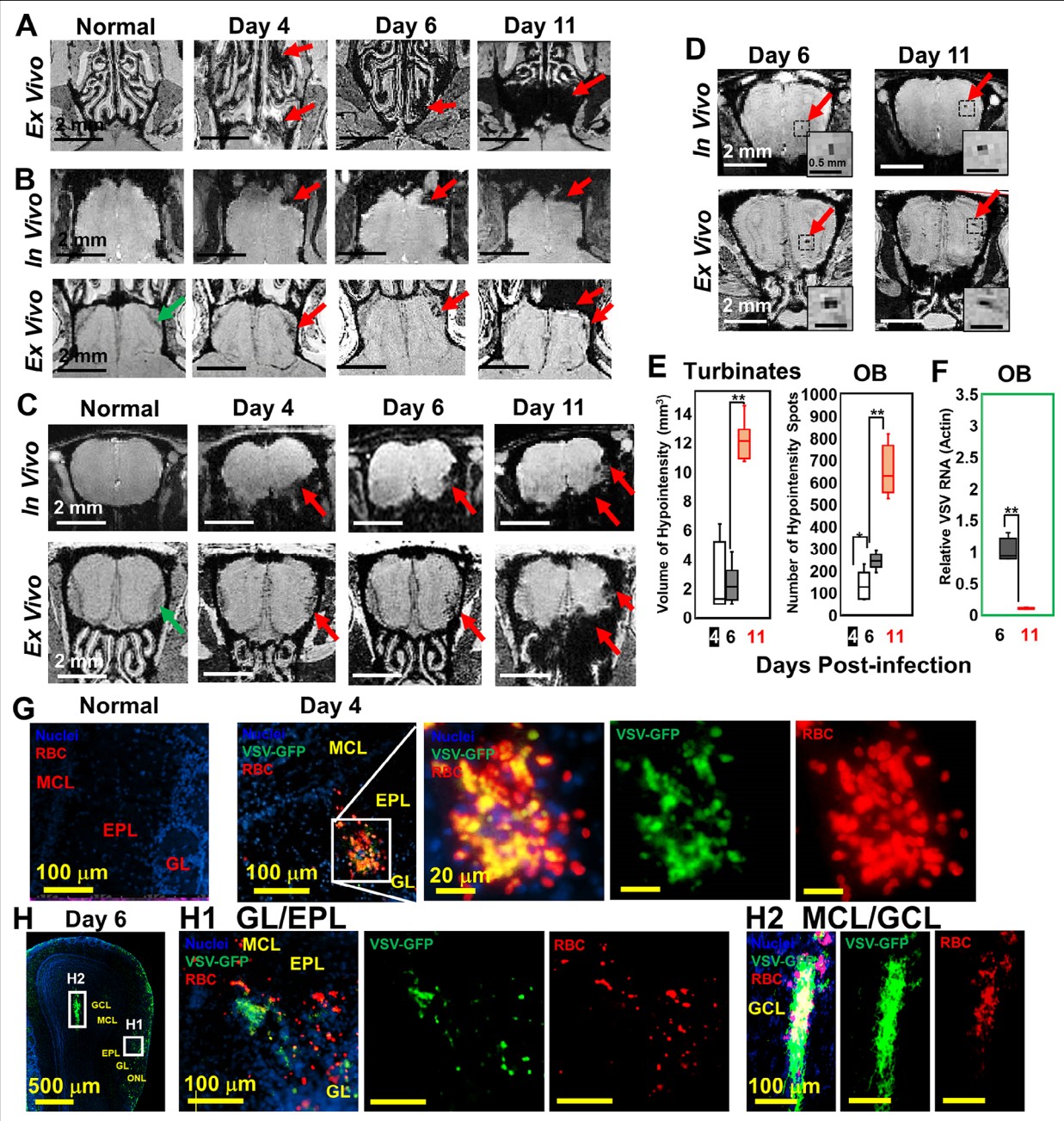

**Figure 1.** Magnetic resonance imaging (MRI) detected microbleeds in turbinates and olfactory bulb (OB) since day 4 post-infection. (**A**) Ex-vivo MRI of turbinates (axial view) from a normal mouse and VSV-infected mice on day 4, 6, and 11 post-infection. (**B**) Axial view and (**C**) coronal view of in-vivo (upper panel) and ex-vivo (lower panel) MRI of OB. The coronal view was sectioned from the bleeding site on the axial view, as indicated by red arrows. (**D**) At the center of the OB, microbleeds were detected since around day 6 by in-vivo MRI. The inserted figure was the enlarged views of the framed bleeding sites in the full views. The scale bar in the full view: 2 mm. The scale bar in the inserted view: 0.5 mm (**D**). (**E**) Quantification of volume of hypointensity in the turbinates and numbers of hypointensity spots in the OB on day 4, 6, and 11 (n=6). * p=0.046; ** p<0.0001. (**F**) Viral titers, represented as relative to actin RNA, at the OB on day 6 and 11 (n=6). ** p<0.0001. (**G, H**) Immunohistochemical (IHC) study to show microbleeds and vesicular stomatitis virus (VSV) in the OB of VSV-infected mice on day 4 (**G**) and 6 (**H**). EPL, external plexiform layer; GCL, granule cell layer; GL, glomerular layer; MCL, mitral cell layer; ONL, olfactory nerve layer. Blue, DAPI for nuclei; Green, VSV-GFP; Red, Alexa Fluor 647-conjugated anti-Ter119 for RBCs.

The online version of this article includes the following figure supplement(s) for figure 1:

**Figure supplement 1.** Magnetic resonance imaging (MRI) detected microbleeds in the turbinate and olfactory bulb (OB) during vesicular stomatitis virus (VSV) infection.

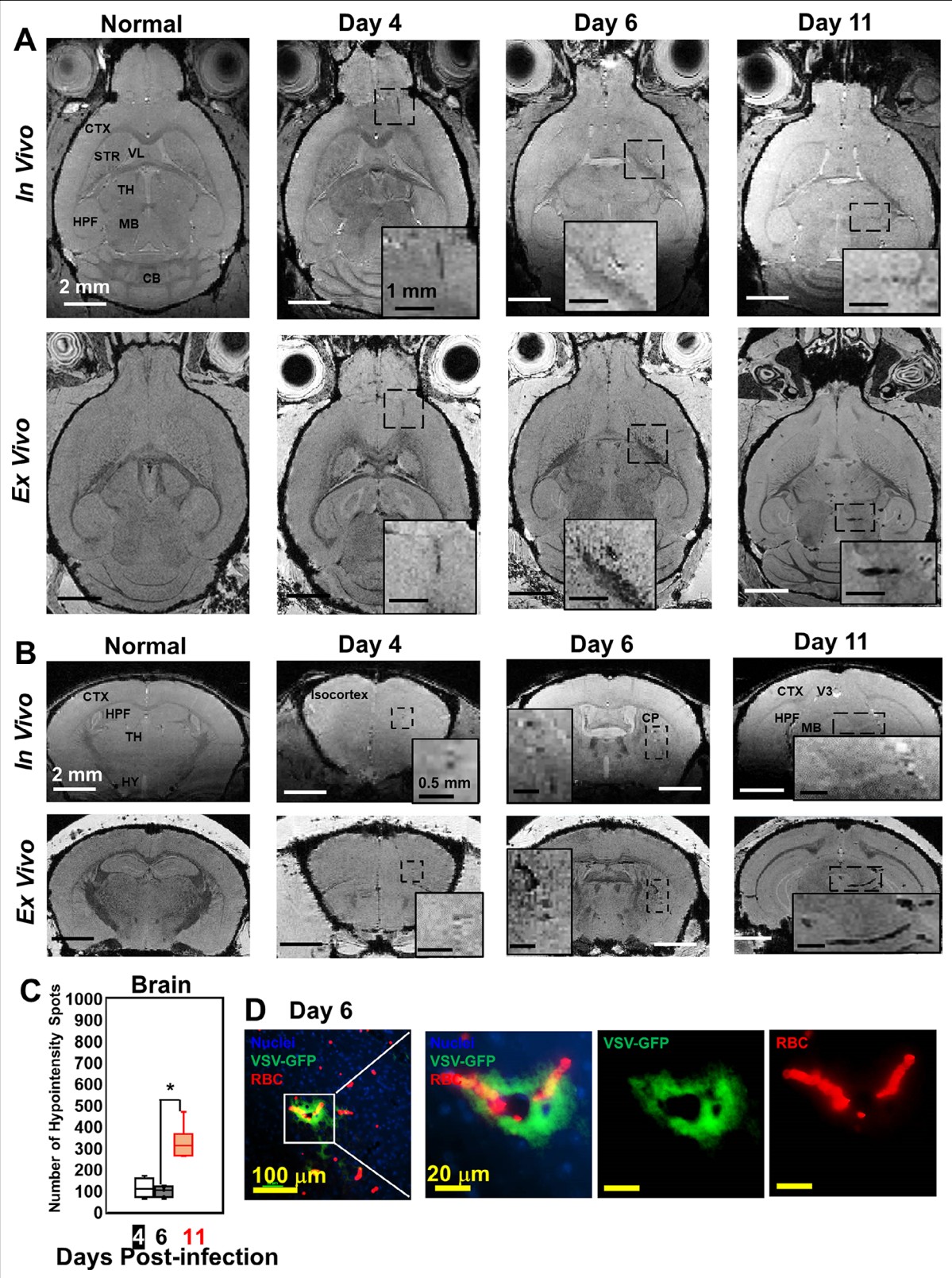

**Figure 2.** Magnetic resonance imaging (MRI) monitored vessel breakdown in the rest of brain. (**A**) Axial view and (**B**) coronal view of in-vivo (upper panel) and ex-vivo (lower panel) MRI of brain, showing the breakdown of vessels from frontal brain to midbrain. The scale bar in the full view: 2 mm. The scale bar in the inserted view: 1 mm (**A**) and 0.5 mm (**B**). CB, cerebellum; CP, caudoputamen; CTX, cerebral cortex; HPF, hippocampal formation; HY, hypothalamus; MB, midbrain; STR, striatum; TH, thalamus; V3, third ventricle; VL, lateral ventricle. (**C**) Quantification of numbers of hypointensity spots in

*Figure 2 continued on next page*

*Figure 2 continued*

the brain (not including OB) on day 4, 6, and 11 (n=6). * p<0.0001. (**D**) Immunohistochemical (IHC) staining to show microbleeds and VSV in the forebrain on day 6. Blue, DAPI for nuclei; Green, VSV-GFP; Red, Alexa Fluor 647-conjugated anti-Ter119 for RBCs.

The online version of this article includes the following figure supplement(s) for figure 2:

**Figure supplement 1.** Magnetic resonance imaging (MRI) monitored vessel breakdown in the rest of brain.

blocking antibodies decreased the number of CD45+ cells by more than 95% on day 6 (*Figure 3B*). Reduced numbers of neutrophils, Ly6C+, CD8+, and CD11b+ cells are shown in *Figure 3—figure supplement 1D*. MRI studies showed that, upon treatment with anti-LFA-1/VLA-4 blocking antibodies, a greater volume of bleeding was detected in the turbinates on day 6, relative to the two control groups, i.e., normal mice treated with anti-LFA-1/VLA-4 antibodies and VSV infected mice treated with isotype control (rat IgG2a) antibodies (*Figure 3C*). Within the OB, upon treatment with blocking antibodies, in-vivo MRI revealed significant bleeding at multiple sites (ONL/GL and the center) as early as day 3 (*Figure 3D and E*). At the ONL/GL, MRI could detect significant bleeding in mice treated with isotype control antibodies on day 4 (*Figure 3D*), which was the same as untreated VSV-infected mice (*Figure 1B*). Blocking immune cell infiltration also increased bleeding at the center of the OB on day 3 (*Figure 3E*), and a large area of hemorrhage was detected in the thalamus on day 6 (*Figure 3F and G*; *Figure 3—figure supplement 1E–G*).

As quantified in *Figure 3H*, the volume of hypointensity in the turbinates increased from 2.1 ± 0.8 mm$^3$ to 5.0 ± 1.0 mm$^3$ on day 6 in anti-LFA-1/VLA-4 treated mice, and the number of bleeding spots increased in the brain by ~2-fold. In addition, blockade of immune infiltration elevated viral RNA in the OB by ~2–3-fold on day 6 (*Figure 3I*). Thus, VSV induces hemorrhage in the brain without involvement of the peripheral immune system. In fact, inhibition of peripheral immune infiltration increased viral RNA, which was associated with more bleeding.

## VSV infects vascular endothelial cells

VSV was previously shown to enter the brain through olfactory sensory neurons via retrograde transneuronal transport and spread in the CNS via both anterograde and retrograde transport (*Beier et al., 2013*; *Cornish et al., 2001*). Although VSV is highly cytopathic, most VSV-infected neurons in the brain are cleared noncytolytically (*Moseman et al., 2020*). While it is generally known that neurons are infected by VSV, we became interested in whether other cell types supported VSV infection. We assessed if cerebrovascular endothelial cells were infected by the virus given the bleeding observed in the brains of VSV-infected mice. As shown in *Figure 4A and B*, we detected VSV in endothelial cells comprising cerebral blood vessels, which sometimes showed evidence of bleeding (*Figure 4C* and quantified in *Figure 4D*). These data suggest that infection of cerebrovascular endothelial cells by VSV may contribute to the mechanism responsible for bleeding.

## Adoptive transfer of antiviral CD8 T cells decreases brain bleeding and increases viral clearance

Because inhibition of peripheral immune infiltration enhanced brain bleeding, we next evaluated whether adoptive transfer of virus-specific CD8 T cells could be used to reduce viral loads and bleeding. 5 × 10$^5$ virus-specific mTomato CD8+ OT-I T cells (purity shown in *Figure 5—figure supplement 1*) activated overnight with ovalbumin (OVA)$_{257–264}$ peptide were transferred i.v. into VSV-OVA infected mice at two time points. For early-stage transfer, OT-I cells were transferred on day 0–2, and MRI was performed on day 6. For peak-stage transfer, OT-I cells were transferred on day 6 and mice were imaged on day 11 (*Figure 5—figure supplement 2*). Adoptive transfer of OT-I cells had a therapeutic benefit in both early- and peak-late-stage paradigms (*Figure 5A–D*). For peak-late-stage treatment, in the turbinates (representative images shown in *Figure 5A*), the volume of bleeding decreased from 12.2 ± 1.3 mm$^3$ to 1.6 ± 1.1 mm$^3$ on day 11 (*Figure 5E*). In the OB (*Figure 5B*), the number of hypointensity spots decreased from 665 ± 127–114 ± 23 on day 11 (*Figure 5F*). In the brain (*Figure 5D*), the number of hypointensity spots decreased from 329 ± 72 to 83 ± 19 on day 11 (*Figure 5G*). T cell transfer reduced microbleeds (*Figure 5B and C*) and also reduced viral RNA in the OB for early stage transfer. Relative VSV to actin RNA decreased from 0.95 ± 0.37 to 0.29 ± 0.07 (*Figure 5H*).

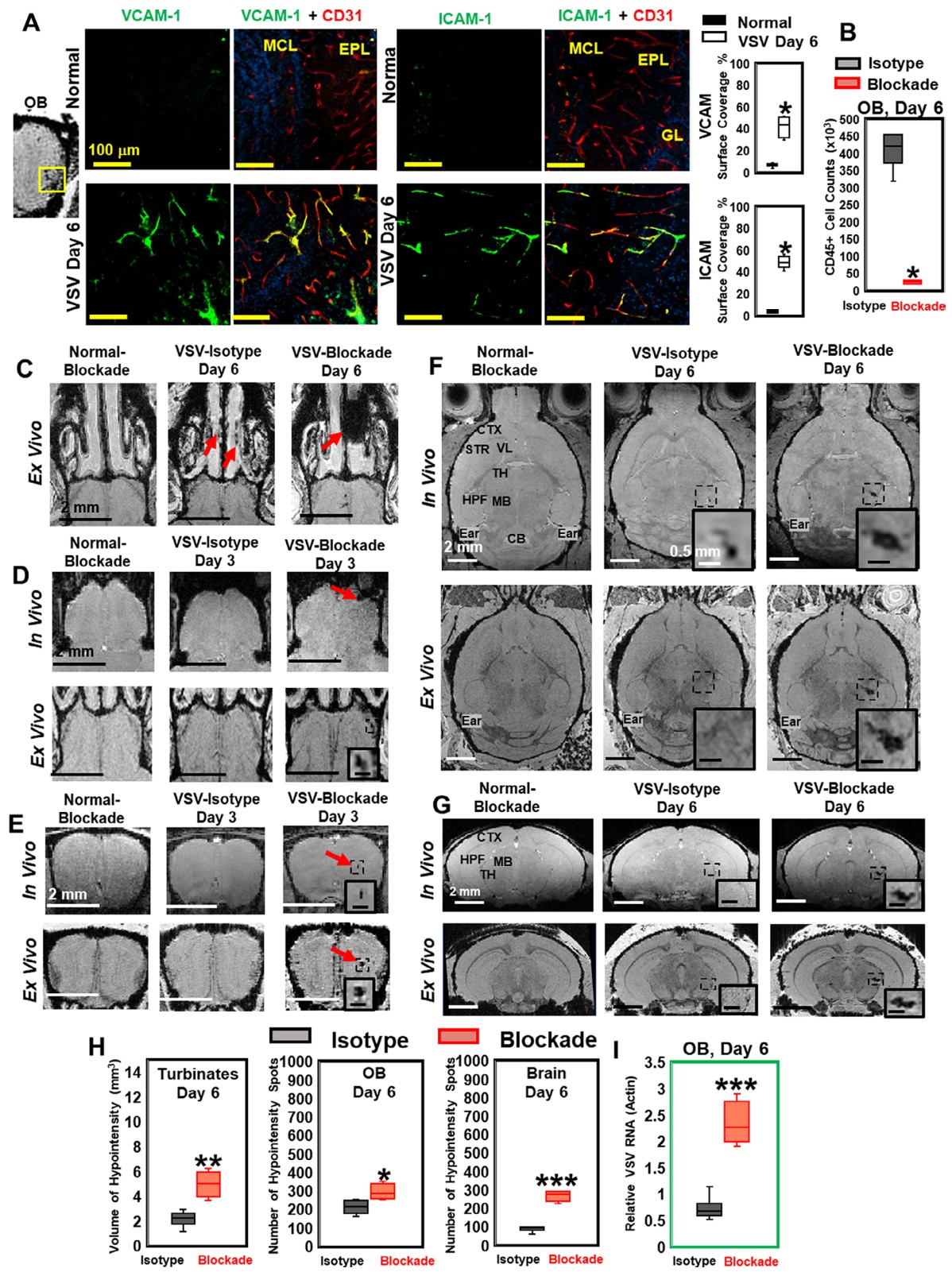

**Figure 3.** Vesicular stomatitis virus (VSV) infection caused vascular endothelium activation and blocking immune cell infiltration by anti-LFA-1/VLA-4 antibodies promoted bleeding in the turbinates, olfactory bulb (OB), and brain. (**A**) Immunohistochemical (IHC) staining showed increased expression of endothelial VCAM-1 and ICAM-1 in the OB on day 6 of infection. n=3 per group. * p<0.0001. (**B**) Anti-LFA-1/VLA-4 blockade antibodies decreased CD45+ cells over 95% in the OB on day 6. n=4 per group. * p<0.0001. (**C**) Treatment with anti-LFA-1/VLA-4 blockade antibodies increased bleeding

*Figure 3 continued on next page*

*Figure 3 continued*

in the turbinates. Ex-vivo MRI of the turbinates (axial view) from normal mouse treated with the blockade antibodies, VSV infected mouse treated with isotype control (rat IgG2a) antibody on day 6, and VSV infected mouse treated with the blockade antibodies on day 6. In the glomerular layer (GL) (**D**) and the granule cell layer (GCL) of OB (**E**), MRI detected bleeding earlier upon treatment with the blockade antibodies. (**F**) Axial and (**G**) coronal view of MRI of the brain showed a large area of hemorrhage at the thalamus of anti-LFA-1/VLA-4 treated mouse on day 6. The coronal view was sectioned from the hemorrhage site on the axial view, as shown in the frame. In (**F**) and (**G**), scale bar in the full view: 2 mm; in the inserted view: 0.5 mm. (**H**) Quantification of the volume of bleeding in the turbinates and the numbers of hypointensity spots in the OB and brain on day 6, in the isotype control group and anti-LFA-1/VLA-4 antibodies treated group (n=4). * p=0.039; ** p=0.005; *** p<0.0001. (**I**) Blocking immune cell infiltration increased viral RNA in the OB on day 6 (n=4). *** p<0.0001.

The online version of this article includes the following figure supplement(s) for figure 3:

**Figure supplement 1.** Vesicular stomatitis virus (VSV) infection caused vascular endothelium activation and blocking immune cell infiltration by anti-LFA-1/VLA-4 antibodies promoted bleeding in the brain.

IHC confirmed that, without T cell transfer, there was a large amount of VSV and microbleeds, and a small number of CD8 T cells at the ONL, GL (*Figure 6A*), and center of the OB on day 6 (*Figure 6D*). *Figure 6A and D* also showed that GL and OB center were major sites of virus invasion and replication, which was consistent with *Figure 1H*. Upon transfer of OT-I cells, an increased number of CD8+ T cells, and a reduced amount of VSV and microbleeds was observed in the GL (*Figure 6B*) and OB center (*Figure 6E*). Quantification of these data is shown *Figure 6C and F*. Some microbleeds,

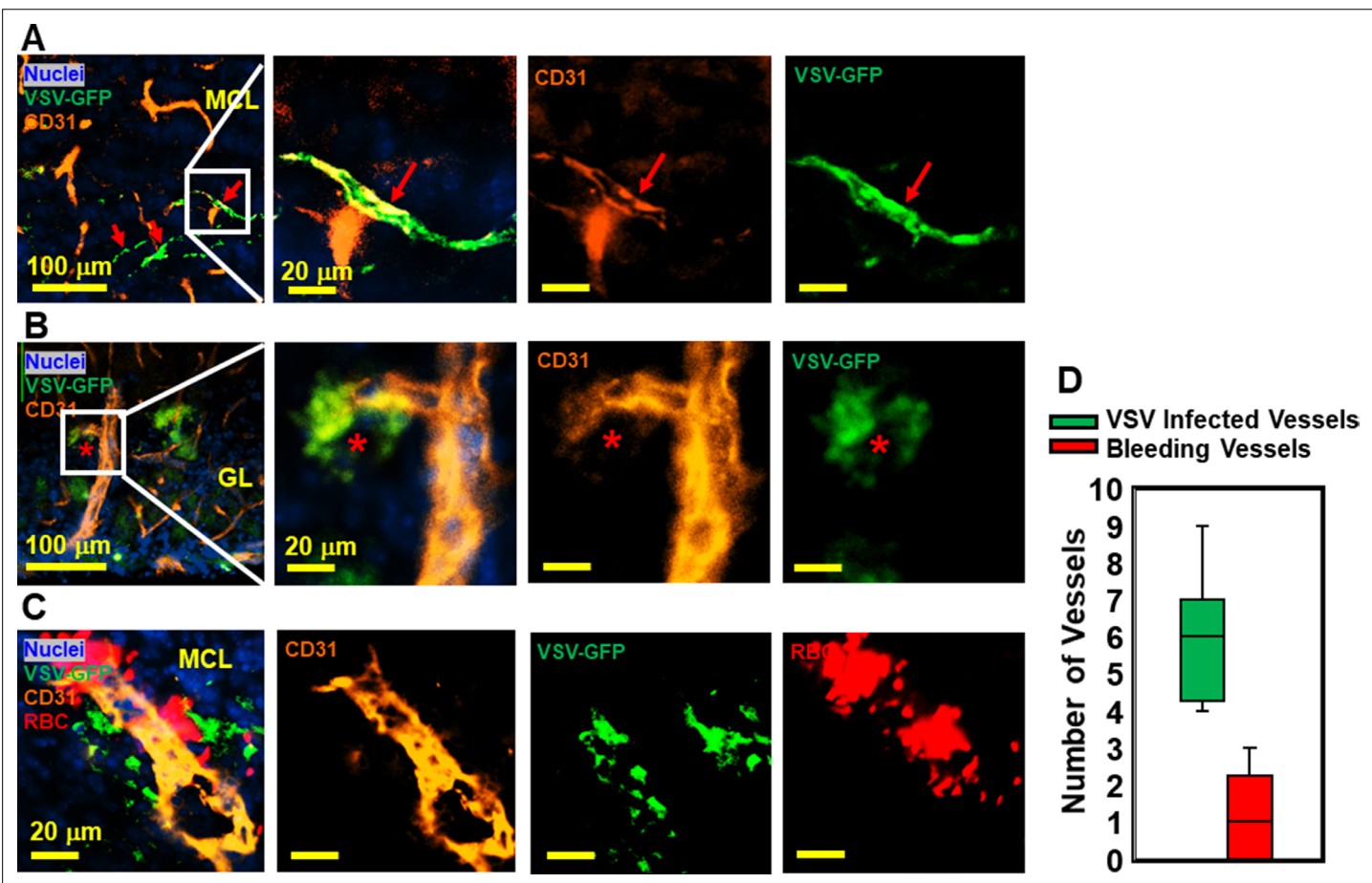

**Figure 4.** Immunohistochemical (IHC) study revealed that vesicular stomatitis virus (VSV) can infect vascular endothelial cells. (**A, B**) Representative views of VSV infected vessels in the mitral cell layer (MCL) and glomerular layer (GL), as indicated by the red arrows and red star. Blue, DAPI for nuclei; Green, VSV-GFP; Orange, Alexa Fluor 647 conjugated anti-CD31. (**C**) Bleedings were observed from VSV infected vessels in the MCL. (**D**) Quantification of vessels colocalized with virus only or colocalized with virus and RBCs in the GL on day 6, under 40 x view (0.2 mm x 0.2 mm). n=3 mice. 6 views were counted manually per mouse. Orange, Alexa Fluor 594 conjugated anti-CD31; Red, Alexa Fluor 700 conjugated anti-Ter119 for RBCs.

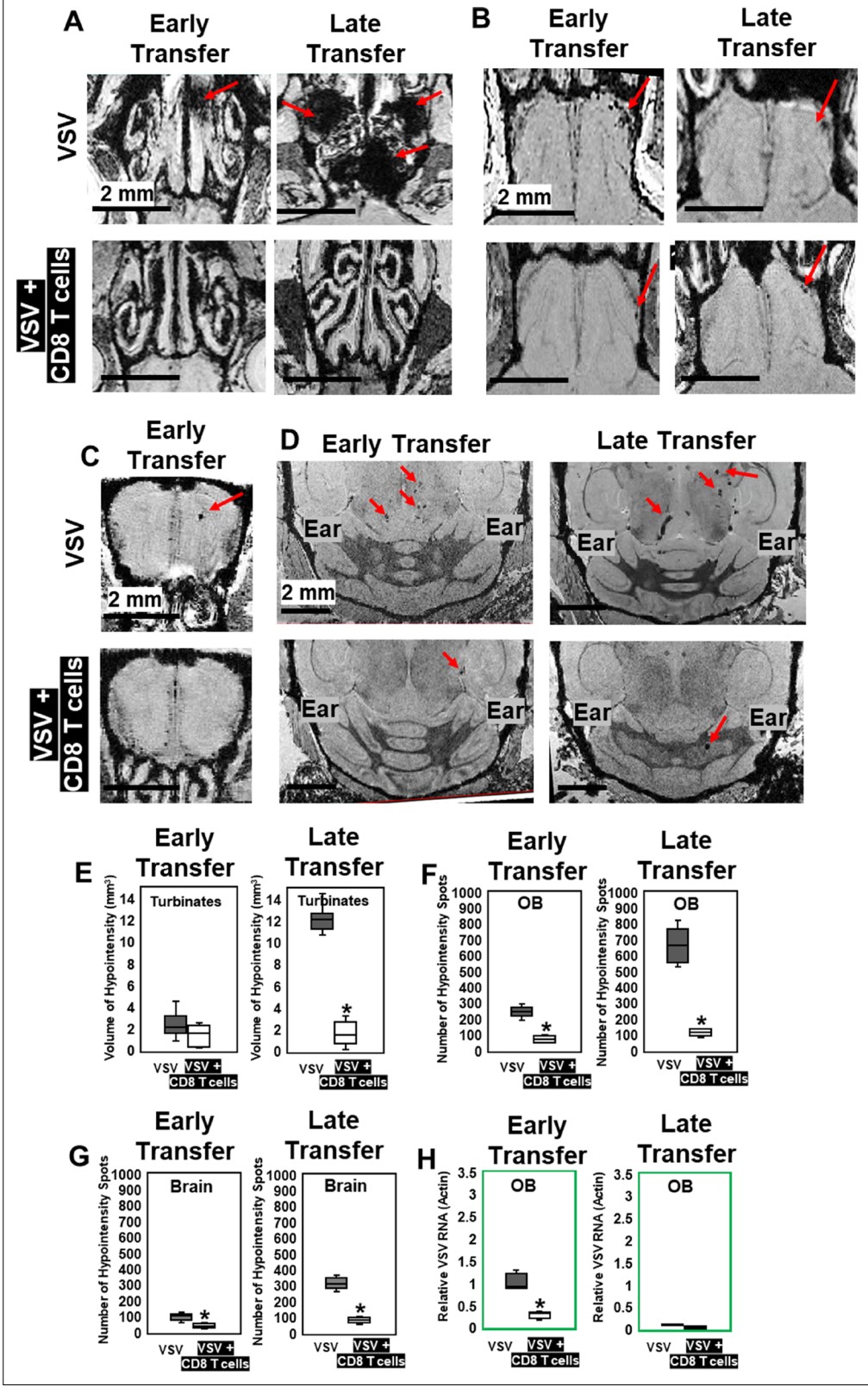

**Figure 5.** MRI study showed that adoptive CD8 T cells transfer reduced cerebrovasculature breakdown. (**A–D**) MRI showed that both early and late transfer of OT-I CD8 T cells reduced bleeds in the turbinates (**A**), GL of the OB (**B**), center of the OB (**C**) and brain (**D**). Red arrows, microbleeds. Quantification of volume of bleeding in the

*Figure 5 continued on next page*

*Figure 5 continued*

turbinates (**E**) and numbers of hypointensity spots in the OB (**F**) and brain (**G**) without or with T cells early and late transfer (n=5–7 mice per group). * p<0.0001. (**H**) CD8 T cells reduced viral RNA on day 6 (n=4). * p<0.0001.

The online version of this article includes the following figure supplement(s) for figure 5:

**Figure supplement 1.** Purity of CD8 T cells used in this study.

**Figure supplement 2.** Experiment paradigms.

however, were still seen in mice receiving antiviral T cells, especially in regions with a high density of T cells.

However, having too many antiviral T cells can also elevate cerebral bleeding. Administering a higher number of virus-specific CD8+ T cells ($3 \times 10^6$ T cells, relative to the therapeutic lower dose $5 \times 10^5$) on day 6 was pathogenic as detected by MRI and IHC (*Figure 6—figure supplement 1*). This finding suggests that antiviral CD8+ T cells at high doses can contribute to vessel damage while attempting to control virus. Thus, care must be used when determining T cell doses for therapeutic adoptive cell transfers.

## Individual MPIO labeled T cells can be detected by MRI

To understand the spatial relation between transferred T cells and microbleeds, we developed a method to label T cells with MPIO particles, which can be detected by MRI with high sensitivity (*Pothayee et al., 2017*; *Shapiro et al., 2004*; *Wu et al., 2006*). Labeling non-phagocytic T cells with contrast agents is challenging. Cationic agents, such as transfection agents and HIV-TAT peptide, or electroporation and mechanoporation have been shown to increase the labeling efficiency (*Bhatnagar et al., 2013*; *Hingorani et al., 2020* ; *Kiru et al., 2022*; *Thu et al., 2012*). A previous study has also shown that T cells can be labeled with MPIOs by way of an antibody-mediated, biotin–streptavidin recognition strategy (*Shapiro et al., 2007*). We therefore modified the surface of streptavidin-MPIO particles with cationic amine groups (*Liu et al., 2012*) to increase the efficiency of uptake by T cells. A representative structure of the particle is shown in *Figure 7A*, and a detailed synthetic scheme is shown in *Figure 7—figure supplement 1*. Briefly, a streptavidin-MPIO was chemically conjugated with ethylenediamine to increase T cell uptake. In addition, anti-CD3 was conjugated to MPIO as an affinity ligand and to increase internalization of the particles by T cells. DyLight 488 was conjugated to MPIO for FACS and fluorescence microscopy studies.

mTomato CD8 + OT-I T cells were incubated with MPIO-DyLight 488 overnight. For FACS analyses, anti-streptavidin-DyLight 650 was used to identify the location of MPIO as either extracellular (i.e. attached to the cell membrane) or intracellular. A detailed flow chart of the FACS method is shown in *Figure 7—figure supplement 2A*. As shown in *Figure 7B* and *Figure 7—figure supplement 2A*, the yield of MPIO-labeled T cells was ~8.3% and the majority of MPIOs were localized inside the cells. Fluorescence images of MPIO labeled OT-I cells in cell culture is shown in *Figure 7—figure supplement 2B*. The mTomato color on the OT-I cells was not bright especially after fixation (*Figure 7—figure supplement 2C*), thus anti-CD8 was used to stain OT-I cells (*Figure 7C*). Electron microscopy (EM) images were used to confirm the cellular location of MPIO (*Figure 7D*, *Figure 7—figure supplement 2D*). 91% MPIOs were localized intracellularly (*Figure 7E*). Some T cells contained more than one MPIO. MPIO-labeled T cells showed MR sensitivity enabling single cell detection (*Figure 7F*).

Labeling of T cells with nanoparticles, especially cell surface labeling, may alter T cell functions (*Stephan et al., 2010*). Several key functions of MPIO-labeled T cells were checked in vitro, including migration (*Figure 7G*) and IL-2 and IFN-gamma release (*Figure 7—figure supplement 2E, F*). There was no significant change in these functions.

## MRI based tracking of T cells and microbleeds on day 1 of infection

With MRI having the sensitivity to detect both single T cells and microbleeds, we used this approach to guide histology studies and determine what gave rise to hypointensities at the earliest stages of VSV infection and inflammation in the brain. We wanted to determine whether vessel breakdown or T cell extravasation occurred first after adoptive transfer of antiviral T cells.

MPIO-labeled OT-I cells were transferred just prior to VSV infection. At 15 hr post-infection, no hypointensities were detected in the OB by MRI (*Figure 8A*). At 24 hr post-infection, the earliest

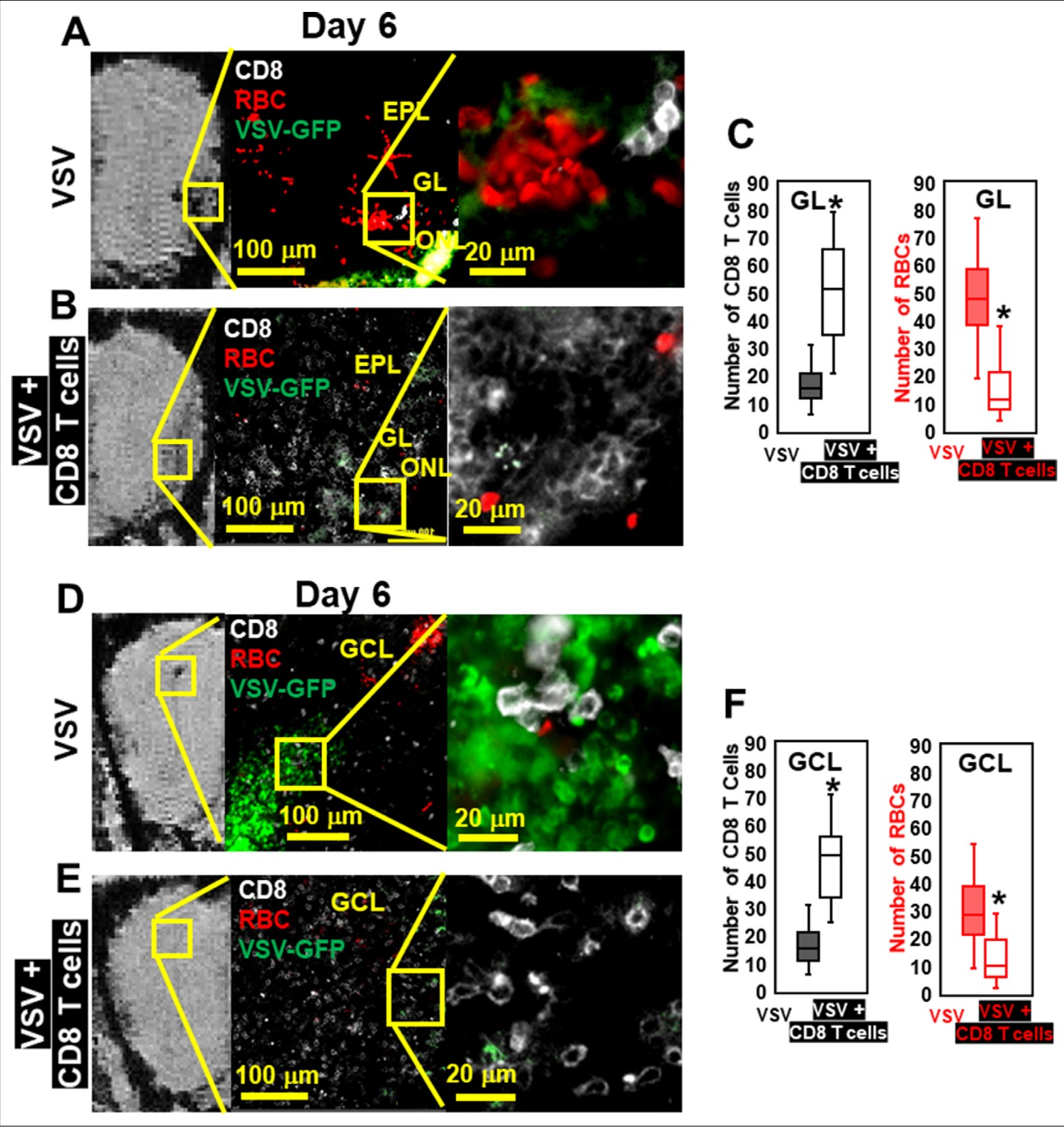

**Figure 6.** Immunohistochemical (IHC) study verified that antiviral CD8 T cells reduced VSV-induced brain bleeding. (**A**) On day 6, VSV-induced bleeding at the olfactory nerve layer (ONL) and glomerular layer (GL) and there was a small number of CD8 T cells. (**B**) CD8 T cells transfer reduced bleeding and virus and there was a larger number of CD8 T cells at the ONL and GL. (**D, E**) In the granule cell layer (GCL), T cell transfer also reduced bleeding and virus comparing with non-T cells transferred. White, Brilliant Violet 421 conjugated anti-CD8 for T cells; Green, VSV-GFP; Red, Alexa Fluor 647 conjugated anti-Ter119 for RBCs. (**C, F**) Quantification of CD8 T cells and bleeds in the GL and GCL, under 40 x view (0.2 mm x 0.2 mm). n=4–6 mice per group. Six views were counted manually per mouse. * p<0.0001.

The online version of this article includes the following figure supplement(s) for figure 6:

*Figure 6 continued on next page*

*Figure 6 continued*

**Figure supplement 1.** Adoptive transfer of overdose of CD8 T cells (3 × 10⁶ T cells) at the peak stage caused bleeding in the olfactory bulb (OB) and brain.

hypointensity spots were detected near the GL (*Figure 8B*) and the center, MCL/GCL areas of the bulb (*Figure 8C*). Similar hypointensities were not detected in the controls infused with unlabeled CD8 T cells. Since MPIOs and blood generate similar T2* effects, IHC was required to assess the cause of the hypointensities detected by MRI. IHC showed that, in the GL, CD8 T cell infiltration and bleeding co-localized and both occurred by 24 hr (*Figure 8D*). The MRI hypointensity in the GL was caused by the combination of MRI contrast from MPIO-labeled T cells and microbleeds. As quantified in *Figure 8H*, 85% of MPIO-labeled T cells were in the region with microbleeds and 15% were in a region with no evidence of bleeding in the ONL/GL. Interestingly, hypointensities in the

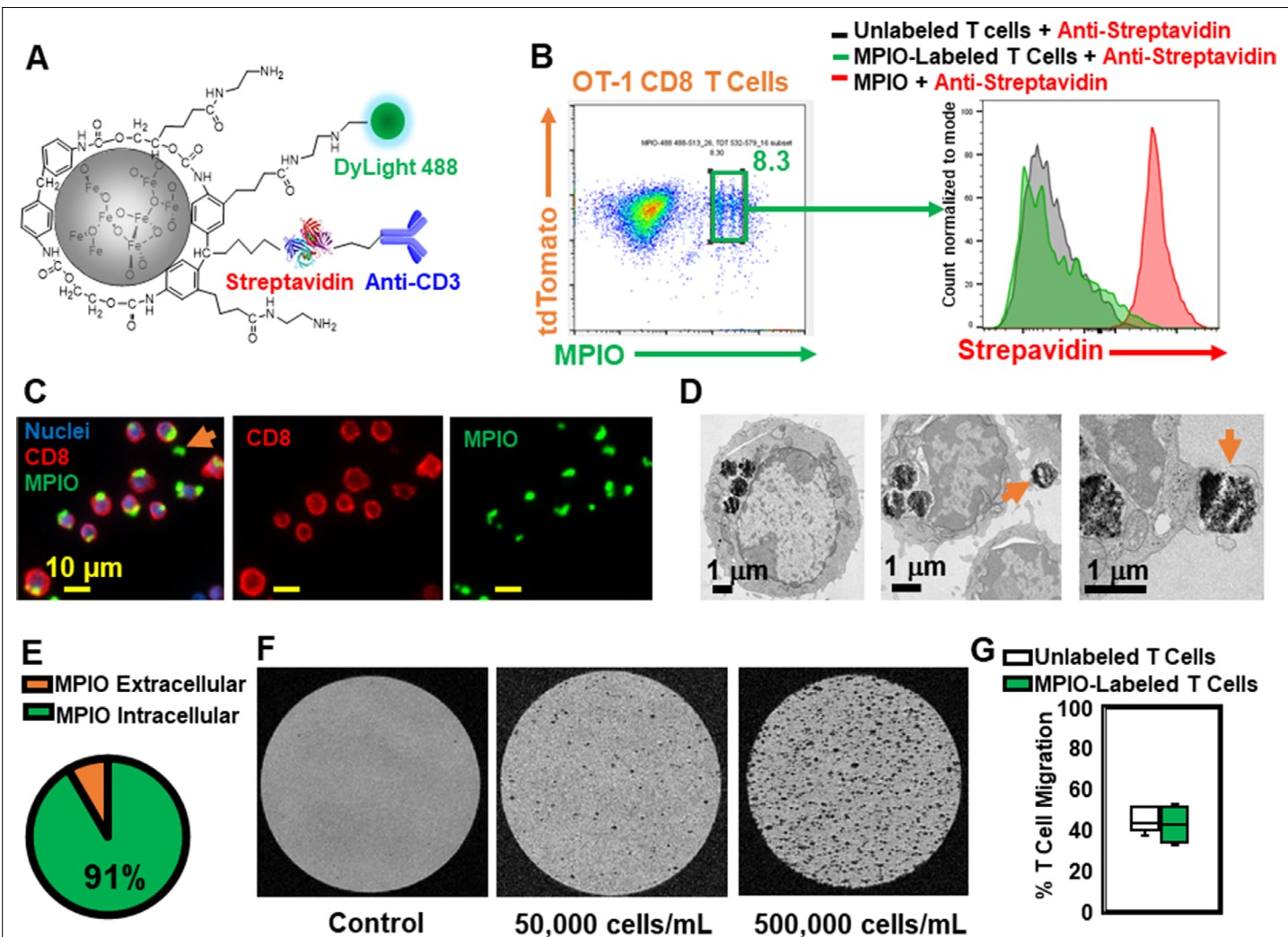

**Figure 7.** Label CD8 T cells with MPIO for Magnetic resonance imaging (MRI) investigation. (**A**) A representative structure of the conjugated MPIO particle used in this study to label T cells. (**B**) FACS MPIO-labeled CD8 T cells, defined as DAPI⁻, mTomato⁺, DyLight 488⁺, DyLight 650⁻. (**C**) Fluorescence images of MPIO-labeled T cells. Green, MPIO-DyLight 488; Red, Alexa Fluor 647 conjugated anti-CD8. (**D**) EM images of CD8 T cells labeled with MPIO intracellularly and extracellularly. (**E**) Quantification of intracellular and extracellular MPIO through EM images. (**F**) MR sensitivity of labeled T cells. (**G**) Percentage of migrating T cells in the migration assay (n=6).

The online version of this article includes the following figure supplement(s) for figure 7:

**Figure supplement 1.** Bioconjugation of MPIO particles to label T cells.

**Figure supplement 2.** Label CD8 T cells with MPIO.

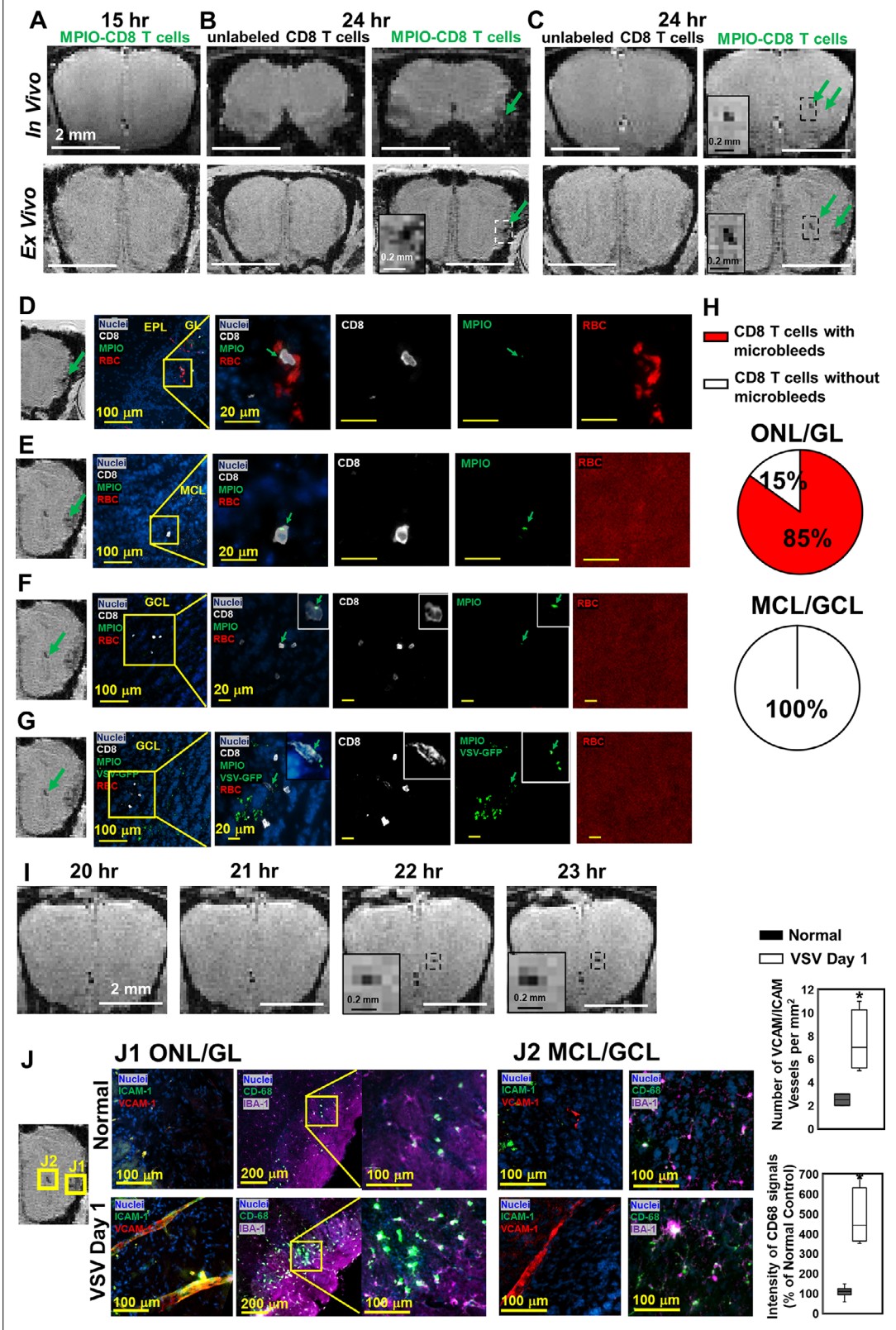

**Figure 8.** Magnetic resonance imaging (MRI) tracking of virus-specific CD8 T cells and detection of brain viral infection on day 1. (**A**) No hypointensity was detected at 15 hr post-infection in the olfactory bulb (OB). (**B, C**) The earliest hypointensity spots were detected near the GL (**B**) and the center, mitral cell layer (MCL) to granule cell layer (GCL) (**C**), at 24 hr post-infection. Similar hypointensities were not detected in the controls infused with

*Figure 8 continued on next page*

*Figure 8 continued*

unlabeled CD8 T cells. RBCs were detected at the GL (**D**), but not near the MCL (**E**) and GCL (**F, G**), as shown by Immunohistochemical (IHC). In (**D–F**), Vesicular stomatitis virus (VSV) with no GFP was used to verify the presence of MPIO-DyLight 488 in the T cells. White, Alexa Fluor 594 conjugated anti-CD8 for T cells. Green, VSV-GFP or MPIO-DyLight 488; Red, Alexa Fluor 700 conjugated anti-Ter119 for RBCs. (**H**) Frequencies of MPIO-labeled T cells present with microbleeds versus without microbleeds at the ONL/GL and MCL/GCL [under 40 x view (0.2 mm x 0.2 mm); n=4 mice; 35 MPIO-labeled T cells total: 26 cells at the ONL/GL and 9 cells at the MCL/GCL]. (**I**) The appearance of MPIO-labeled T cells at the center at 22 hr post-infection. (**J**) Activation of vessel endothelium cells and microglia were found at the ONL/GL (**J1**) and MCL/GCL (**J2**) on day 1. n=3 per group. 3–6 views were quantified per mouse. * p<0.0001.

The online version of this article includes the following figure supplement(s) for figure 8:

**Figure supplement 1.** Time-lapse magnetic resonance imaging (MRI) showing the possible movement of MPIO-labeled T cells at the center of olfactory bulb (OB) from 22–24 hr-post-infection.

**Figure supplement 2.** MPIO-labeled T cells increased magnetic resonance imaging (MRI) sensitivity to microbleeds.

MCL/GCL areas were due only to MPIO-labeled T cells, and no evidence for bleeding was detected (*Figure 8E–G*). To determine if MRI based tracking of T cells could determine the precise time of T cell arrival into the MCL/GCL area and whether these cells migrated to this area from within the bulb, time-lapse MRI was performed from 20–24 hr post-infection. The appearance of hypointensity, caused by MPIO-labeled T cells was detected at 22 hr (*Figure 8I*, *Videos 1 and 2*). No migration tracks of T cells from ONL/GL, where bleeding had already begun, to the center from MRI was detected. Thus, T cells at the center of the bulb likely extravasated from the local vasculature. Representative MRI from 22–24 hr post-infection is shown in *Figure 8—figure supplement 1*. No evidence for T cell migration over this 2 hr time interval could be detected on the MRI distance scale.

In *Figure 8D–F*, to verify the presence of MPIO-DyLight 488 particles in transferred T cells, VSV with no GFP was used in the experiments. In *Figure 8G*, to track the virus, VSV-GFP was used. Virus localized to the center of the bulb on day 1. As shown in *Figures 1H and 6D*, the center of the bulb was a major site of virus infection. It is likely that the ability of T cells to traffic to sites of viral infection rapidly prior to vascular damage explains why adoptive transfer of T cells greatly reduced brain bleeding during VSV infection. That T cells can enter sites of viral infection prior to vessel break down is consistent with earlier studies using the LCMV model to study this issue on the surface of the brain (*Herz et al., 2015*).

On day 1, at the locations that CD8 T cells and microbleeds were detected, activation of vascular endothelium cells (as indicated by VCAM-1 and ICAM-1) and microglia (as indicated by CD68) were also detected by IHC (*Figure 8J*). Thus, MRI of microbleeds and T cell tracking provides a noninvasive method for early detection of cerebrovascular inflammation.

The combination of MRI contrast from MPIO-labeled T cells and bleeds increased MRI sensitivity to areas where there was both microbleeds and T cell infiltration (*Figure 8B*). To test whether this enabled better detection of sites of inflammation at later stages of infection, MRI hypointensity spots were counted after T cell transfer at day 6 and 11 (*Figure 8—figure supplement 2*). More hypointensity

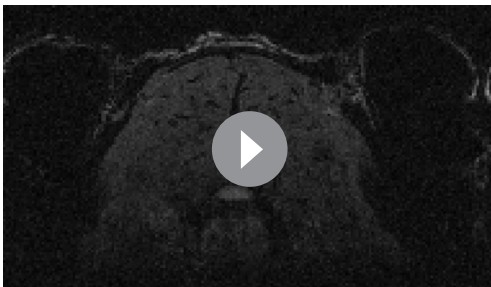

**Video 1.** Coronal views of the whole brain at 21 hr post-infection, before the appearance of the hypointensity.

https://elifesciences.org/articles/74462/figures#video1

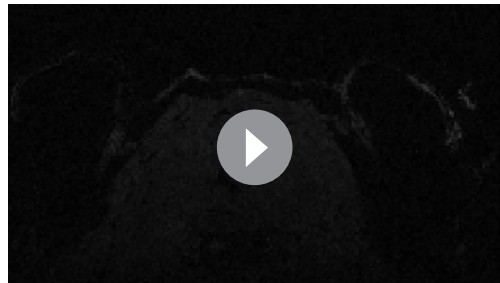

**Video 2.** Coronal views of the whole brain at 22 hr post-infection, after the appearance of the hypointensity.

https://elifesciences.org/articles/74462/figures#video2

spots could be detected likely due to additive effects of bleeding and T cell detection. The limitation at this stage is that we cannot distinguish microbleeds from MPIO-labeled T cells, and some of the MRI sites may be due to bleeding alone or T cell infiltration alone. However, the total signal from MPIO-labeled immune cells plus bleeds made sites of inflammation more readily detectable by MRI. This provides an approach for imaging neuroinflammation when peripheral cells enter the brain. This was shown in previous studies focused on stroke and traumatic brain injury (*Drieu et al., 2020*; *Foley et al., 2009*) where large numbers of peripheral immune cells infiltrate. The present study has been able to detect small bleeds and very few T cells infiltrating in an antigen specific manner.

## Discussion

Inflammatory processes are involved in most neurological disorders but are especially important during viral infections. Interactions between the immune system and CNS as well as the pathways by which immune cells infiltrate the brain differ depending on the anatomical location, structure of the barriers, and type of disorder. Using microbleeds as a marker of cerebrovasculature breakdown, we studied the relationship between vascular disruption and CD8 T cell infiltration using T2*-weighted MRI during a neurotropic VSV infection of the mouse brain.

$T2^*$-weighted MRI exhibits a high sensitivity to microbleeds. This high sensitivity is due to paramagnetic properties of deoxy-hemoglobin from RBCs or its degradation product, hemosiderin (*Haller et al., 2018*). Paramagnetic and supermagnetic materials create a magnetic field gradient that is detected by T2*-weighted MRI and has an amplifying affect because the field gradients can extend at least 50 times greater than the actual size of the material (*Lauterbur et al., 1996*). This amplification is critical for MRI detection of vascular effects during brain activation (BOLD functional MRI), for sensitive detection of bleeding by MRI, and for many pre-clinical studies that have labeled and imaged cells with MRI. Sensitivity down ~1 pg iron per voxel can be achieved at high resolution (*Shapiro et al., 2004*). This is why we followed in-vivo MRI with ex-vivo MRI of fixed brains because ex-vivo MRI can provide significantly higher resolution to guide histology. One RBC contains ~0.2 pg iron as calculated from *Ozment and Turi, 2009*. Thus, there is the potential to detect ~5–10 fully deoxygenated RBCs, which facilitates high sensitivity visualization of vessel bleeds where only a small amount of blood has entered the brain. Thus, MRI was shown to be more sensitive than CT to detect bleeding in human brain. For example, it was demonstrated in a study of herpes simplex virus induced encephalitis that T2*-weighted MRI could detect bleeds in the brain within 2 days during the acute phases of the disease, whereas 1–2 weeks were required for CT-based detection (*Gupta et al., 2012*). The presence of microbleeds in patients with stroke and traumatic brain injury is increasingly recognized as an indicator of worse outcome (*Griffin et al., 2019*; *Haller et al., 2018*). Iron deposition can initiate continued neuroinflammation and brain damage. Indeed, MRI detected brain vessel damage and microbleeds in COVID patients was recently used to guide histological studies and uncover mechanisms of inflammation associated with SARS-CoV-2 (*Lee et al., 2021*; *Riggle et al., 2020*; *Poyiadji et al., 2020*).

Peripheral pathology and inflammation associated with viral infection can contribute to brain bleeding without viral infection of the brain. Previous studies showed that following i.n. inoculation, VSV replicated in the mouse lung, caused a transient viremia, and was found transiently in other organs, including liver and spleen. The virus was cleared from the periphery (lung, plasma, and liver) within 2–3 days post-infection (*Publicover et al., 2006*; *Ramsburg et al., 2005*; *Roberts et al., 1999*). In the CNS, viral titer (*Moseman et al., 2020*) and histology (*Figures 1, 2 and 8*) showed that VSV infected brain very efficiently, peaking on day 6 before being cleared by day 8. Microbleeds largely co-localized with sites of VSV replication in the brain. Based on these results, we think that direct CNS infection, rather than peripheral pathology, causes brain microbleeds in this model. However, no attempt was made to rule out peripheral infection to brain bleeding in the present study.

Guided by MRI, we demonstrated that neurotropic VSV can cause brain vessel breakdown without the involvement of the peripheral immune system (*Figure 3*). VSV spike G-protein binds to low-density lipoprotein receptor (LDL-R) and enters cells via receptor-mediated endocytosis (*Nikolic et al., 2018*). It was reported that endothelium of brain capillaries express LDL-R (*Méresse et al., 1989*). In addition, a cell culture experiment showed that VSV-GFP infected and lysed endothelial cells (*Patel et al., 2020*). This may explain the presence of VSV along vessels and provides a potential mechanism for viral disruption of vascular integrity independent of peripheral immune cell infiltration into brain.

There is also microglia and astrocyte activation during VSV infection that could potentially lead to vessel inflammation. Future studies are required to reveal the exact mechanism by which VSV infects endothelial cells and the role this plays in causing damage to cerebral blood vessels.

The present study demonstrated that the peripheral immune response to VSV infection is important for the maintenance of vascular integrity and pathogen clearance, as blockade of CNS immune cell entry via administration of anti-LFA1/VLA4 antibodies increased brain bleeding and viral RNA (*Figure 3*). Moreover, supplementing the immune response by adoptively transferring activated virus-specific CD8+ T cells at either early (day 0–2) or later (day 6) time points post-infection protected cerebral vessels. Nevertheless, some bleeding was still observed even after transfer of CD8+ T cells at days 0–2, and CD8+ T cells responding to the virus were associated with vessel damage at this early stage. The transferred CD8+ T cells were not effective at completely blocking early viral entry from nasal olfactory sensory neurons into the OB. Notably, at both early and late stages of infection, adoptive transfer of MPIO-labeled virus-specific CD8+ T cells at the proper dose protected against brain bleeding (*Figures 1E and 2C* versus *Figure 8—figure supplement 2C, D*). Therefore, MPIO-labeled CD8+ T cells retained their therapeutic functions, indicating that the MPIO labeling did not have a negative effect on T cell functionality in-vivo. This is consistent with the in-vitro assays of T cell function after MPIO labeling that revealed no impact on T cell migration or cytokine production.

VSV showed tropism for the neurons in the GL and GCL and transferred virus-specific CD8+ T cells first arrived in these two layers between 15 and 24 hr post-infection. A previous study demonstrated that CD8+ T cells controlled neuronal virus in this model through engagements with cross-presenting microglia (*Moseman et al., 2020*). In our study, adoptive transfer of virus-specific CD8+ T cells improved CNS viral control leading to reduced viral RNA and preserving vascular integrity in the GL and GCL (*Figure 5*). These two layers of the bulb contain proliferating immature neurons in rodents (*Batista-Brito et al., 2008*; *Lois and Alvarez-Buylla, 1994*). These immature neurons are generated by precursor cells in the subventricular zone (SVZ) and traffic through the rostral migratory stream (RMS) to different layers of the OB, primarily the GL and GCL, where they differentiate into neurons (*Batista-Brito et al., 2008*; *Lois and Alvarez-Buylla, 1994*). Many viruses have a tropism for immature neurons (*Li et al., 2016*), and the presence of immature neurons in these two layers may in part explain the tropism of VSV.

Based on our MRI cell tracking experiment, the appearance of hypointensity at 22 hr (*Figure 8I*) is specific for OB inflammation. We do not observe hypointensity in any other brain locations (*Videos 1 and 2*). We evaluated the possible movement of MPIO-labeled T cells by time-lapse MRI (*Mori et al., 2014*; *Figure 8—figure supplement 1*), but migration of T cells from the ONL/GL region, which had early signs of bleeding, was not detected. Furthermore, there was no change in the location of T cells from 22–24 hr after cells arrived in the MCL/GCL area. There are two possibilities to explain our inability to detect T cell movement over a 2 hr time interval: (1) the T cells under investigation may have been attached to blood vessels and required more time to extravasate, or (2) although T cell velocities in the CNS have been clocked at ~10 µm/min (*Herz et al., 2015*), their paths are often tortuous and influenced by antigen presenting cells displaying cognate peptide MHC as well as local chemokine gradients. Thus, upon entering a site of viral infection, the labeled T cells may not have traveled far enough in 2 hr for us to detect their movement by MRI. We did not image mice beyond 24 hr post-infection due to the possibility of bleeding. Nevertheless, future studies will monitor T cell movement over longer time intervals.

In conclusion, our MRI guided studies have revealed that VSV has the ability to cause vascular damage even at the early stages of infection. This damage is worse if peripheral immune cell infiltration is blocked and can be reduced by adoptively transferring antiviral T cells. It is clear that control of a CNS viral infection requires maintenance of a precise balance between reducing viral titer and preserving tissue integrity, since transfer of too many T cells led to more extensive bleeding (*Figure 6—figure supplement 1*). This is consistent with previous studies showing that CD8+ T cells can also contribute to cerebrovascular pathology following *Plasmodium berghei* infection (*Swanson et al., 2016*). Identifying specific viral and immune mechanisms that result in cerebrovascular damage is a clinically relevant area of research, and our studies demonstrate that MRI can be used to evaluate the extent of pathology after a CNS viral infection as well as the efficacy of a therapeutic intervention like adoptive antiviral T cell transfer. In future studies, it will be important to develop more sensitive MRI contrast agents that can be distinguished from blood (*Barbic et al., 2021*; *Zabow et al., 2014*)

and / or to use quantitative T2* analysis at different magnetic field strengths. Development of new MRI based approaches that can distinguish blood from iron-labeled immune cells will aid studies designed to assess the relationship between vascular breakdown and the peripheral inflammatory response. Finally, study of pathogen-specific T cells by MRI at the level of single cell detection should advance our understanding T cell entry and movement throughout the brain, which could aid in optimizing adoptive cell therapies for the treatment of different infections and cancers.

# Materials and methods

## Key resources table

| Reagent type (species) or resource | Designation | Source or reference | Identifiers | Additional information |
|---|---|---|---|---|
| Cell line (*Mus musculus*) | C57BL/6-Tg(TcraTcrb)1,100Mjb/J (OT-I) mice | The Jackson Laboratory | Cat#: 003831 | |
| Cell line (*Mus musculus*) | B6.129(Cg)-Gt(ROSA)26Sortm4(ACTB-tdTomato,-EGFP)Luo/J (mTomato) | The Jackson Laboratory | Cat#: 007676 | |
| Antibody | Brilliant Violet 421 conjugated anti-CD8 (Clone 53–6.7) (Rat monocolonal) | BioLegend | Cat# 100,753 | (1:100 dilution) (2 µg/mL) for IHC |
| Antibody | Alexa Fluor 594 conjugated anti-CD8 (Clone 53–6.7) (Rat monocolonal) | BioLegend | Cat# 100,758 | (1:100 dilution) (2 µg/mL) for IHC |
| Antibody | Alexa Fluor 700 conjugated anti-CD8 (Clone 53–6.7) (Rat monocolonal) | BioLegend | Cat# 100,730 | (1:80 dilution) ≤0.25 µg per $10^6$ cells in 100 µl volume for flow cytometry |
| Antibody | Brilliant Violet 421 conjugated anti-CD45 (Clone 30-F11) (Rat monocolonal) | BioLegend | Cat# 103,134 | (1:80 dilution) ≤0.25 µg per $10^6$ cells in 100 µl volume for flow cytometry |
| Antibody | APC conjugated anti-Thy1.2 (Clone 30-H12) (Rat monocolonal) | BioLegend | Cat# 105,312 | (1:80 dilution) ≤0.25 µg per $10^6$ cells in 100 µl volume for flow cytometry |
| Antibody | Brilliant Violet 605 conjugated anti-CD11b (Clone M1/70) (Rat monocolonal) | BioLegend | Cat# 101,257 | (1:80 dilution) ≤0.25 µg per $10^6$ cells in 100 µl volume for flow cytometry |
| Antibody | Brilliant Violet 785 conjugated anti-Ly6C (Clone HK1.4) (Rat monocolonal) | BioLegend | Cat# 128,041 | (1:80 dilution) ≤0.25 µg per $10^6$ cells in 100 µl volume for flow cytometry |
| Antibody | PE conjugated anti-Ly6G (Clone 1A8) (Rat monocolonal) | BioLegend | Cat# 127,608 | (1:80 dilution) ≤0.25 µg per $10^6$ cells in 100 µl volume for flow cytometry |
| Antibody | Alexa Fluor 594 conjugated anti-CD31 (Clone 390) (Rat monocolonal) | BioLegend | Cat# 102,432 | (1:250 dilution) (2 µg/mL) for IHC |
| Antibody | Alexa Fluor 647 conjugated anti-CD31 (Clone 390) (Rat monocolonal) | BioLegend | Cat# 102416 | (1:250 dilution) (2 µg/mL) for IHC |
| Antibody | Alexa Fluor 647 conjugated anti-TER-119 (Rat monocolonal) | BioLegend | Cat# 116,218 | (1:200 dilution) (2.5 µg/mL) for IHC |
| Antibody | Alexa Fluor 647 conjugated anti-TER-119 (Rat monocolonal) | BioLegend | Cat# 116,220 | (1:200 dilution) (2.5 µg/mL) for IHC |
| Antibody | anti-IBA-1 (Rabbit polycolonal) | FUJIFILM Wako | Distributor Barcode No 019–19741 4987481428584 | (1:500 dilution) (0.4 µg/mL) for IHC |
| Antibody | anti-CD68 antibody (Clone FA-11) (Rat monocolonal) | Bio-Rad | Cat# MCA1957GA | (1:200 dilution) (5 µg/mL) for IHC |
| Antibody | Alexa Fluor 647 conjugated anti- CD106/VCAM-1 (Clone 429 or MVCAM.A) (Rat monocolonal) | BioLegend | Cat# 105,712 | (1:250 dilution) (2 µg/mL) for IHC |
| Antibody | Alexa Fluor 488 anti-CD54/ICAM-1 (Clone YN1/1.7.4) (Rat monocolonal) | BioLegend | Cat# 116,112 | (1:250 dilution) (2 µg/mL) for IHC |
| Antibody | Anti-LFA-1 (Clone M17/4) (Rat monocolonal) | BioCell | Cat# BE0006 | (500 µg each IP injection) |

*Continued on next page*

*Continued*

| Reagent type (species) or resource | Designation | Source or reference | Identifiers | Additional information |
|---|---|---|---|---|
| Antibody | anti-VLA-4 (Clone PS/2) (Rat monocolonal) | BioCell | Cat# BE0071 | (500 µg each IP injection) |
| Antibody | rat IgG2a (Rat monocolonal) | BioCell | Cat# BE0089 | (1000 µg each IP injection) |
| Antibody | APC conjugated anti-IFN-γ (Clone XMG1.2) (Rat monocolonal) | BioLegend | Cat# 505,809 | (1:20 dilution) ≤1 µg per $10^6$ cells in 100 µl volume for flow cytometry |
| Antibody | PE/Cy7 conjugated anti-IL-2 (Clone JES6-5H4) (Rat monocolonal) | BioLegend | Cat# 503,831 | (1:160 dilution) ≤0.125 µg per $10^6$ cells in 100 µl volume for flow cytometry |
| Antibody | anti-Streptavidin (Rabbit polycolonal) | Thermo Fisher Scientific | Cat# S6390 | (20 µl for $10^7$ T cells for FACS) |
| Antibody | biotin anti-CD3 (Clone 145–2 C11) (Rat monocolonal) | BioLegend | Cat# 100,304 | 25 µl (12.5 µg) for $5 \times 10^9$ MPIO particles during conjugation |
| Commercial assay or kit | Dynabeads Untouched mouse cd8 cells kit | Invitrogen | Cat# 11,417D | |
| Commercial assay or kit | QCM chemotaxis cell migration assay, 24-well (3 µm) | Millipore Sigma | Cat# ECM505 | |
| Commercial assay or kit | Cyto-Fast Fix/Perm buffer set | BioLegend | Cat# 426,803 | |
| Commercial assay or kit | miRNeasy mini kit | Qiagen | Cat# 217,084 | |
| Commercial assay or kit | iScript cDNA synthesis kit | Bio-Rad | Cat# 1708890 | |
| Commercial assay or kit | SYBR Green PCR master mix | Applied Biosystems | Cat# 1708890 | |
| Chemical compound, drug | DyLight 488-NHS | Thermo Fisher Scientific | Cat# 46,403 | |
| Chemical compound, drug | DyLight 650-NHS | Thermo Fisher Scientific | Cat# 62,266 | |
| Chemical compound, drug | recombinant murine JE/MCP-1 (CCL2) | Peprotech | Cat# 250–10 | 10 ng/mL for migration assay |
| Chemical compound, drug | $OVA_{257-264}$ peptide | Sigma Aldrich | Cat# S7951 | 2 µg/ml for OT-I cell culture |
| Chemical compound, drug | 2-Mercaptoethanol | Sigma Aldrich | Cat# M3148 | |
| Chemical compound, drug | 2-(N-morpholino)ethanesulfonic acid | Sigma Aldrich | Cat# M3885 | |
| Chemical compound, drug | 1-Ethyl-3-[3-dimethylaminopropyl]carbodiimide hydrochloride | Sigma Aldrich | Cat# E6383 | |
| Chemical compound, drug | sulfo-N-hydroxysuccinimide | Sigma Aldrich | Cat# 130,672 | |
| Chemical compound, drug | Ethylenediamine | Sigma Aldrich | Cat# E26266-100ML | |
| Chemical compound, drug | brefeldin A | BioLegend | Cat# 420,601 | |
| Software, algorithm | MIPAV | http://mipav.cit.nih.gov | | |
| Other | DAPI stain | Invitrogen | Cat# D1306 | (1 µg/mL) |
| Other | flow cytometry staining buffer | Invitrogen | Cat# 00-4222-26 | |
| Other | Dynabeads MyOne Streptavidin C1 | Thermo Fisher Scientific | Cat# 65,001 | |

## Animal procedures

C57BL/6 J (B6), B6.129(Cg)-Gt(ROSA)26Sortm4(ACTB-tdTomato,-EGFP)Luo/J (mTomato), and C57BL/6-Tg(TcraTcrb)1,100Mjb/J (OT-I) mice were purchased from The Jackson Laboratory. mTomato x OT-I mice were bred and maintained under specific pathogen-free conditions at the National Institutes of Health. All mice in this study were handled in accordance with the Institute of Laboratory

Research guidelines and were approved by the Animal Care and Use Committee of the National Institute of Neurological Disorders and Stroke.

## Intranasal (i.n.) VSV Infection

Twelve- to 13-week-old C57BL/6 mice were used in all the infection and control experiments. Recombinant VSV expressed a truncated form of OVA and with or without GFP. For intranasal (i.n.) VSV-OVA infection, a dose of $3.5 \times 10^4$ PFU in 10 µL of PBS was pipetted into the opening of each nostril.

## MRI study

For MRI scans, VSV infected mice were placed into the chamber shown in *Figure 5—figure supplement 2*. MRI experiments were performed at 11.7T on a Bruker Biospec MRI system (Bruker BioSpin). T2*-weighted 3D gradient-recalled echo (GRE) sequences were used for acquisitions. For in-vivo imaging, the following parameters were used: isotropic resolution = 75 µm, TE/TR = 10/30ms, FA = 10°, NA = 3, scan time = 36 m. For time-lapse MRI: isotropic resolution = 75 µm, TE/TR = 8/25ms, FA = 10°, NA = 1, each scan time = 10m2 s208ms. After in-vivo acquisitions, mice were perfused transcardially with 5% formalin. The heads were post-fixed with 10% formalin. 24 hr before ex-vivo MRI imaging acquisition, heads were transferred to PBS buffer. For ex-vivo MRI: isotropic resolution = 50 µm, TE/TR = 20/40ms, FA = 15°, NA = 12, scan time = 15h8 m.

## Quantification of bleeding in the nasal turbinates and brain

Ex-vivo high-resolution MRI was used to quantify nasal and brain bleeding in VSV infected mice. Because bleeding in the turbinates was so extensive, volumes were measured. Hypointensity volumes were obtained by manually drawn serial voxels of interest (VOI) using the Medical Image Processing, Analysis, and Visualization (MIPAV) program (http://mipav.cit.nih.gov) (*Saar et al., 2015*). Hypointensities in the brain were more focal so the number of spots was counted manually. The volume of hypointensities and number of hypointensity spots from normal healthy mice were used as background and subtracted from the measurements for each infected mouse.

## Quantitative real-time PCR (RT-qPCR) for VSV RNA

Total RNA was isolated from OB using miRNeasy Mini Kit (Qiagen). Subsequently, 1 µg of RNA was treated with amplification grade DNAse I (Life Technologies) to remove any contaminating genomic DNA. We used 250 ng of total RNA for cDNA synthesis using iScript cDNA Synthesis Kit (Bio-Rad). All Q-PCRs were performed in 20 µL volumes using SYBR Green PCR master mix (Applied Biosystems) in a 96-well optic tray using a CFX96 Real-Time PCR machine (Bio-Rad). The reactions were conducted in duplicate, and samples without reverse transcriptase were used as non-template controls. Forward primer (specific for the VSV genes): TGA TGA TGC ATG ATC CAG CTC T; reverse primer: ACA CAC CTC CAA TGG AAG GGT. Actin RNA was used as internal standard. Q-PCR reactions were run with an initial denaturation temperature of 95 °C for 10 min, followed by 40 cycles of three-step amplification (denaturation at 95 °C for 10 s, 60 °C annealing for 15 s and extension at 72 °C for 20 s).

## Adhesion molecule blockade studies

Anti-LFA-1 (clone M17/4), anti-VLA-4 (clone PS/2), and isotype control (rat IgG2a) antibodies were purchased from BioXcell. Mice were treated with a combination of 500 µg anti-LFA-1 and 500 µg anti-VLA-4, or 1000 µg isotype control antibody at day 1, 3, and 5 post-infection through intraperitoneal injection. Normal healthy mice were used as the control group.

To evaluate the efficacy of antibody blockades, cell counting and flow cytometry analysis were performed. On day 6 of infection, mice were perfused with cold PBS and olfactory bulbs were isolated. Tissues were dissected with scissors and digested in 1 mg/mL Collagenase-D and 0.25 mg/mL DNAse in RPMI Complete Medium (Gibco) for 30 min at 37 °C with frequent trituration. Tissues were then dissociated into single cell suspensions by smashing and passed through a 100 µm filter mesh. Cells were then spun down at 350 g for 6 min and resuspended in 3 mL of a 37% isotonic Percoll solution. The samples were spun at 350 x g for 6 min, and the supernatant was removed. Pelleted cells were washed with cold RPMI and spun down. Absolute cell numbers were counted before staining. Cells were suspended in Fc block solution for 10 min. A 2 x antibody cocktail was added directly to cells for 20 min before a final wash and resuspension in FACS buffer for flow cytometry analysis. Antibodies

were from BioLegend: CD45 (BV421; 30-F11), anti-Thy1.2 (APC, 30-H12), anti-CD8 (AF700; 53–6.7), anti-CD11b (BV605; M1/70), anti-Ly6C (BB790; HK1.4), anti-Ly6G (PE; 1A8).

## Adoptive transfer of CD8 T cells for MRI

We considered day 6 post-infection as the peak of encephalitis in this model of VSV infection (*Huney-cutt et al., 1994*). For MRI studies ranging from day 1–6 post-infection, we injected $5 \times 10^5$ mTomato OT-I CD8+ T cells into the tail vein on day 0 or 2. T cells were injected intravenously on day 6 for MRI studies performed on day 7–11. *Figure 5—figure supplement 2* provides an illustration of these two experiment paradigms.

## OT-I CD8 T cell isolation, culture, and labeling with MPIO

Splenocytes were isolated from mTomato OT-I mice. Splenocytes were prepared by smashing with the textured end of a sterile 10 mL syringe followed by pipetting through a 40 micron filter into a 15 mL conical tube. The cells were collected by centrifugation at 300 g for 6 min and decanted. Cell pellets were resuspended in mouse ACK lysis buffer and waited for 10 min at room temperature followed by two washes with PBS (w/1% FBS). OT-I CD8 T cells were isolated using a Dynabeads Untouched Mouse CD8 Cells Kit (Invitrogen). The purity of CD8 T cells was confirmed by flow cytometry (*Figure 5—figure supplement 1*), using the Beckman Coulter's MoFlo Astrios cell sorter and analyzed by the Summit 6.3.1 software. CD8 T cells were cultured overnight in RPMI medium supplemented with $OVA_{257-264}$ peptide (2 µg/ml), 2-mercaptoethanol (50 µM), FBS (10%), penicillin (50 U/ml), streptomycin (50 µg/ml), and l-glutamine (2 mM) and incubated at 37 °C with 5% $CO_2$ in a humidified (85% humidity) cell culture incubator.

For MPIO labeling, mTomato OT-I CD8+ T cells were incubated with MPIO at a ratio of 1:5 overnight. The CD8 T cells were then washed with cell staining buffer (BioLegend), harvested, and stained with anti-streptavidin-DyLight 650 and DAPI for 30 min on ice. Anti-streptavidin-DyLight 650 was used to identify the cellular location of MPIO, since MPIO was modified with streptavidin. MPIO labeled OT-I cells, defined as DAPI⁻, mTomato⁺, DyLight 488⁺, DyLight 650⁻, were collected. We used the Beckman Coulter's MoFlo Astrios cell sorter installed with the Summit 6.3.1 software for cell sorting.

## MPIO bioconjugation for T cell labeling

The synthetic scheme is shown in the *Figure 7—figure supplement 1*. MPIO particles with streptavidin modification and carboxylic acid group on the surface (Dynabeads MyOne Streptavidin C1, Thermo Fisher Scientific) was used for the bioconjugation reactions. 0.5 ml of MPIO particles ($5 \times 10^9$) was resuspended in 10 ml of 2-(N-morpholino) ethanesulfonic acid buffer (MES, 50 mM, pH 5.5). 1-Ethyl-3-[3-dimethylaminopropyl] carbodiimide hydrochloride (EDAC) and sulfo-N-hydroxysuccinimide (Sulfo-NHS) were added to the particle suspension. The final concentrations of EDAC and NHS were 75 mM and 150 mM, respectively. The reaction mixture was stirred at room temperature for 30 min. The resulting NHS ester was purified by a magnetic stand separator (Miltenyi Biotec). 20 µl of ethylenediamine (shall be large folds excess of calculated carboxyl groups on the particles) was dissolved in 10 ml PBS and pH was adjusted to 7.5. The purified NHS ester was added to the ethylenediamine solution. The reaction mixture was allowed to stir for 4 hr at room temperature. The resulted MPIO-$NH_2$ particles were purified by a magnetic separator and washed 2 times with borate buffer (50 mM, pH 8.5, with 0.05% Tween 20). 50 µg of DyLight 488 or DyLight 650-NHS ester (Thermo Fisher Scientific) was added to MPIO-$NH_2$ particles in 5 ml of borate buffer. Covered the vial with aluminum foil to prevent photobleaching and placed it on a slowly rotating platform for 1 hr at room temperature. Wash the beads 3 times with antibody washing and binding buffer (AWBB, 50 mM Tris-HCl, 150 mM NaCl, 0.05% Tween 20, pH 8.0). 25 µl Biotin anti-mouse CD3 stock solution (12.5 µg, BioLegend) was added to 5 ml of the beads in AWBB. Covered the tube with aluminum foil and placed it on the rotating platform for 30 min at room temperature. The resulted particles were washed and kept in AWBB. Right before T cell labeling, beads were washed three times with RPMI medium.

## Electron microscopy

Freshly sorted cells were spun at 350 x g for 5 min. 1 mL of 4% glutaraldehyde in 0.1 M cacodylate buffer at pH 7.4 was added gently to the cell pellet without disturbing the pellet. Then one small drop of 20% bovine albumin was added to the solution using a fine tip pipette. After 1 hr of fixation at

room temperature, the sample was stored in refrigerator overnight. The pellet was treated with 1% OsO4 plus 1% ferrocyanide in cacodylate buffer for 1 hr and 1% uranyl acetate in acetate buffer at pH 5.0 overnight, dehydrated in a series of ethanol and embedded in epoxy resins. Thin sections were counterstained with uranyl acetate and lead citrate and examined with a JEOL 1200 EXII transmission electron microscope. Images were photographed with a bottom-mounted digital CCD camera (AMT XR-100, Danvers, MA, USA).

### Histology

Following perfusion or ex-vivo MRI acquisition, brains were isolated and post-fixed with 5% formalin for 2 hr. Brains were then transferred into 15% sucrose for 1 day, followed by 30% sucrose for 1 day. Brains were cut coronally and embedded in tissue freezing medium (Fisher HealthCare). Cryosections were cut coronally (30 µm) by a cryostat (Leica CM 1850). Brain sections were stained using a standard procedure for free-floating immunohistochemistry. Briefly, brain sections were rinsed with TBS 3 × 5 min, followed by blocking in 5% normal serum in TBS with 0.5% Triton X-100 for 2 hr at 4 °C. Then the sections were stained in the following antibody solutions in TBS with 3% normal serum and 0.3% Triton X-100 overnight at 4 °C. To label red blood cells (RBC), Alexa Fluor 647 or Alexa Fluor 700 conjugated anti-TER119 (2.5 µg/mL; BioLegend) was used. For vascular endothelial cells, Alexa Fluor 594 or Alexa Fluor 647 conjugated anti-mouse CD31 (2 µg/mL; clone 390; BioLegend) was used. To study vessel inflammation, Alexa Fluor 488 anti-CD54 antibody (2 µg/mL, ICAM-1, YN1/1.7.4, BioLegend), and Alexa Fluor 647 anti-CD106 antibody (2 µg/mL, VCAM-1, 429 MVCAM.A, BioLegend) were used. For CD8 T cells, Brilliant Violet 421 or Alexa Fluor 594 conjugated anti-mouse CD8a (2 µg/mL; clone 53–6.7; BioLegend) was used. To study microglia and activation, rabbit anti-IBA-1 antibody (0.4 µg/mL, FUJIFILM Wako) and rat anti-CD68 antibody (5 µg/mL, clone FA-11, Bio-Rad) was used. For secondary antibody staining, Alexa Fluor 700 goat anti-rabbit or Alexa Fluor 488 goat anti-rat were used, and the staining was at 4 °C for 2 hr. Then tissue sections were put on the Superfrost microscope slides (Fisherbrand). After dried overnight at room temperature, slides with brain sections were then coverslipped using ProLong Gold antifade reagent with or without DAPI (Thermo Fisher Scientific). Images were captured using a Nikon Eclipse Ti microscope (Nikon).

### Quantification of vessel VCAM-1 and ICAM-1 coverage and intensity of CD68 signals

Images obtained under 20 x view (0.4 mm x 0.4 mm) were used for quantification. ImageJ was used for quantification. During quantifications of vessel coverage, the CD31 area was taken as 100% and the VCAM-1 or ICAM-1 positive area was expressed as a percentage normalized to CD31 area.

### Migration assay

T cell migration assays were performed using QCM Chemotaxis Cell Migration assay, 24-well (3 µm) (Millipore Sigma), according to supplier's instruction. $1 \times 10^5$ unlabeled or MPIO-labeled CD8 T cells, in serum free and chemoattractant free RPMI, were plated in the upper wells. In the lower well, 10% FBS and 10 ng/mL T cell chemoattractant MCP-1/CCL-2 were added to the RPMI. After 16 hr, T cells that migrated into the lower chamber were counted by Trypan Blue assay.

### IL-2 and IFN-gamma release assay for T cell function

Briefly, MPIO-labeled OT-1 CD8 T cells or unlabeled CD8 T cells ($1 \times 10^6$) were treated with brefeldin A (BioLegend) for 4 hr. Then cells were stained with APC anti-mouse IFN-γand PE/Cy7 anti-mouse IL-2 antibodies (BioLegend), by using the Cyto-Fast Fix/Perm Buffer Set (BioLegend) followed the Intracellular Flow Cytometry Staining Protocol from BioLegend.

### Statistical analysis

Statistical analysis was carried out with the Student's t-test. A $p<0.05$ was considered statistically significant.

### Acknowledgements

This research was supported by the intramural program of the National Institute of Neurological Disorders and Stroke (NINDS), National Institutes of Health (NIH). The authors would like to thank Dr. Susan Cheng for the excellent EM work and Dr. Jasmin Herz for the valuable discussion during the development of T cell labeling and tracking method.

## Additional information

### Funding

| Funder | Grant reference number | Author |
|---|---|---|
| the Intramural Program of the National Institute of Neurological Disorders and Stroke | 1ZIANS003047-15 | Alan P Koretsky |
| the Intramural Program of the National Institute of Neurological Disorders and Stroke | 1ZIANS003112-13 | Dorian B McGavern |

The funders had no role in study design, data collection and interpretation, or the decision to submit the work for publication.

### Author contributions

Li Liu, Conceptualization, Investigation, Methodology, Supervision, Writing - original draft, Writing - review and editing; Steve Dodd, Ryan D Hunt, Nikorn Pothayee, Tatjana Atanasijevic, Nadia Bouraoud, Dragan Maric, E Ashley Moseman, Selamawit Gossa, Investigation, Methodology; Dorian B McGavern, Conceptualization, Investigation, Supervision, Writing - review and editing; Alan P Koretsky, Conceptualization, Investigation, Methodology, Supervision, Writing - review and editing

### Author ORCIDs

Li Liu 
Alan P Koretsky 

### Decision letter and Author response

Decision letter https://doi.org/10.7554/eLife.74462.sa1
Author response https://doi.org/10.7554/eLife.74462.sa2

## Additional files

### Supplementary files

• Transparent reporting form

### Data availability

The source data of this study wilb be available in Dryad (https://doi.org/10.5061/dryad.79cnp5hwp).

The following dataset was generated:

| Author(s) | Year | Dataset title | Dataset URL | Database and Identifier |
|---|---|---|---|---|
| Liu L, Dodd S, Hunt R, Pothayee N, Atanasijevic T, Bouraoud N, Maric D, Moseman E, Gossa S, McGavern D, Koretsky A | 2021 | Dataset for: Early detection of cerebrovascular pathology and protective antiviral immunity by MRI | https://dx.doi.org/10.5061/dryad.79cnp5hwp | Dryad Digital Repository, 10.5061/dryad.79cnp5hwp |

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
