## [Editor Report]

This manuscript is of broad interest to researchers studying central nervous system (CNS) infections and associated pathology. Utilizing a rodent intranasal infection model as a route of CNS entry, novel magnetic resonance imaging (MRI) approaches are explored to non-invasively track sites of brain vascular microbleeds and the associated immune cell invasion, particularly virus-specific CD8^+^ T cells and their relationship to cerebrovascular pathology in living brains. This study is a timely report about how MRI can detect a potential biomarker for early detection of CNS infections and is used as a surrogate readout for treatment efficacy. It will be of interest to research communities involved in immunotherapies across a broad disease continuum as well as imaging physicists.

---

## [Decision Letter]

**Decision letter after peer review:**

Thank you for submitting your article "Early detection of cerebrovascular pathology and protective antiviral immunity by MRI" for consideration by *eLife*. Your article has been reviewed by 3 peer reviewers, and the evaluation has been overseen by a Reviewing Editor and Jeannie Chin as the Senior Editor. The following individual involved in review of your submission has agreed to reveal their identity: Erik Shapiro (Reviewer #1).

Essential revisions:

1. Demonstrate capacity to track virus specific T cells and cerebrovascular pathology in real time in living brains; and show impact on peripheral pathology.

2. Anti-LFA/VLA-4 blockade used to block peripheral immune cell CNS infiltration revealed that microbleeds were detected earlier and covered larger volumes, which was consistent with impaired viral control. However, the efficacy of the antibody blockade should be confirmed experimentally and noted in results.

Comment/Data/Modify:

1. Figure 1F – it is stated that the peak in viral loads in the OB is at day 6 however no earlier time points are shown. It would be informative to measure viral loads at day 4 post infection.

2. Figure 3 – it is important to confirm anti-LFA-1/VLA-4 actually blocked immune cell influx into the brain. Flow cytometry quantitating the absolute number of lymphocytes in the brains with and without treatment would strengthen the results.

3. Please provide additional quantitative analyses for Figure 4 and 8 on hyperintensities and colocalization with virus or RBC only (Figure 4) and relative frequencies of magnetically labelled T cells present in a pre-vascular compromised region versus in the microbleed, as measured by MRI and histology (Figure 8).

4. Figure 5 – the labelling on this figure is a little confusing (day 6 = early transfer and day 11 late transfer). It would be useful if the authors simplify the labelling to early and late transfer.

5. Figure 7a – to demonstrate binding specificity of the streptavidin can the authors overlay the histogram of the MPIO- OT-I cells in the culture.

6. Figure S9 – The data in this figure is very interesting (T cell dose impacts microbleeding) however the first time Figure S9 brought up is in the discussion – this data needs to be either removed or relocated into the Results section.

7. Removing sup Figure 5; The supplementary figures are not presented in chronological order – this should be amended so they are presented in the order they are introduced in the text.

8. – Lines 81-82: The depiction of "early stage" may not be fitting is this context since T2* changes for microbleeds will persist and are not stageable.

– Line 127 indicates microbleeds were already evident in the midbrain beginning at d4 and line 130 again notes microbleeds were observed in the midbrain at days 8-11. The significance of these findings should be stated more clearly.

– Line 167, Figure 3H depicts viral RNA , not virus titers.

9. Though this is a different antibody, magnetic labeling of T cells by antibody linked MPIOs has been accomplished by Medford et al., and should be acknowledged in the manuscript: https://pubmed.ncbi.nlm.nih.gov/17541955/.

10. Figure 6c – validation of this data using a more robust and quantitative read out (flow cytometry of brain tissue measuring OT-I cells numbers) would be useful. Also it is unclear why anti-CD8 was used instead of dTomato to identify the transferred OT-I cells. It would be interesting to distinguish the localisation of the endogenous CD8 and activated OT-I cells in these sections.

11. Please provide a control experiment using transfer of labelled CD8 T cells with heterologous specificity (e.g. LCMV, EBV specific) to assess if antigen specificity is required for CD8 T cell signal? This approach also address concerns regarding active migration vs passive accumulation of cells.

You should include a reference for 1 pg detection limit for iron per voxel by mri.

Is there a reference for the ability to detect 10 RBCs by MRI?

The magnetic susceptibility generated by deoxygenated blood is orders of magnitude below that of MPIOs. Is there no way reduce the TE in the MRI experiment to substantially reduce the signal from the microbleeds and keep the signal of the magnetically labeled cells? Or to additionally perform T2-weighted imaging to minimize the signal from the MPIOs?

These points have been suggested. It would be great if data are available but the editorial group understand that they may be beyond the scope of the submission. Please however provide a comment or justification in the revised version and response to reviewers:

9. Figure 4 – it is suggested based on the IHC data that VSV infects vascular endothelial cells. To strengthen this claim, I recommend the authors purify out CD31+ endothelial cells and measure VSV RNA in these cells. [may not be achievable]

*Reviewer #1 (Recommendations for the authors):*

Though this is a different antibody, magnetic labeling of T cells by antibody linked MPIOs has been accomplished by Medford et al., and should be acknowledged in the manuscript: https://pubmed.ncbi.nlm.nih.gov/17541955/.

You should include a reference for 1 pg detection limit for iron per voxel by mri.

Is there a reference for the ability to detect 10 RBCs by MRI?

The magnetic susceptibility generated by deoxygenated blood is orders of magnitude below that of MPIOs. Is there no way reduce the TE in the MRI experiment to substantially reduce the signal from the microbleeds and keep the signal of the magnetically labeled cells? Or to additionally perform T2-weighted imaging to minimize the signal from the MPIOs?

*Reviewer #2 (Recommendations for the authors):*

1. More rigorous investigation of potential cerebrovascular microbleeds following systemic or respiratory infections not affecting the CNS would provide a more definitive evidence that direct CNS infection, rather than peripheral pathology, is the causation of microbleeds.

2. Experimental: Anti-LFA/VLA-4 blockade used to block peripheral immune cell CNS infiltration revealed that microbleeds were detected earlier and covered larger volumes, which was consistent with impaired viral control. However, the efficacy of the antibody blockade should be confirmed experimentally and noted in results.

*Reviewer #3 (Recommendations for the authors):*

1. This is an interesting paper, and the capacity to track virus specific T cells and cerebrovascular pathology in real time in living brains would be a major technological advance. However, the model proposed by the authors does not achieve this. It was necessary in this system to perform IHC on fixed brain sections to differentiate T cell influx and microbleeds

2. Figure 1F – it is stated that the peak in viral loads in the OB is at day 6 however no earlier time points are shown. It would be informative to measure viral loads at day 4 post infection.

3. Figure 3 – it is important to confirm anti-LFA-1/VLA-4 actually blocked immune cell influx into the brain. Flow cytometry quantitating the absolute number of lymphocytes in the brains with and without treatment would strengthen the results.

4. Figure 4 – it is suggested based on the IHC data that VSV infects vascular endothelial cells. To strengthen this claim, I recommend the authors purify out CD31+ endothelial cells and measure VSV RNA in these cells.

5. Figure 5 – the labelling on this figure is a little confusing (day 6 = early transfer and day 11 late transfer). It would be useful if the authors simplify the labelling to early and late transfer.

6. Figure 6c – validation of this data using a more robust and quantitative read out (flow cytometry of brain tissue measuring OT-I cells numbers) would be useful. Also it is unclear why anti-CD8 was used instead of dTomato to identify the transferred OT-I cells. It would be interesting to distinguish the localisation of the endogenous CD8 and activated OT-I cells in these sections.

7. Figure 7a – to demonstrate binding specificity of the streptavidin can the authors overlay the histogram of the MPIO- OT-I cells in the culture.

8. Figure S9 – The data in this figure is very interesting (T cell dose impacts microbleeding) however the first time Figure S9 brought up is in the discussion – this data needs to be either removed or relocated into the Results section.

---

## [Author Response]

Essential revisions:1. Demonstrate capacity to track virus specific T cells and cerebrovascular pathology in real time in living brains; and show impact on peripheral pathology.

We appreciate this comment. To address this concern, we conducted new experiments. In these experiments, MRI was used to detect the earliest virus specific T cell infiltrating the olfactory bulb and to assess whether T cells got to this region prior to bleeding. We found no evidence for T cell migration from the edge of the bulb where bleeding had already begun (Figure 8). The new Figure 8I shows the appearance of a MPIO-labeled T cell in the brain at 22 hr-post infection. Thus, we can show the moment a T cell enters the brain. Figures 1-6 showed the capacity of T2*-weighted high-resolution MRI to detect small microbleeds as an early form of cerebrovascular pathology associated with viral entry into the brain. These new data show that T cells can enter infected regions prior to bleeding, likely explaining the therapeutic effect of viral specific T cells on preventing brain bleeding in this model.

We did not directly address whether peripheral pathology associated with viral infection contributes to brain bleeding. VSV was inoculated into the nose, and previous studies demonstrated that the virus is cleared from most peripheral tissues in 2-3 days and there is minimal evidence of peripheral pathology (Ramsburg et al., 2005; Roberts et al., 1999) https://pubmed.ncbi.nlm.nih.gov/16809308/, https://doi.org/10.1128/Jvi.73.5. 3723-3732, https://doi.org/10.1128/Jvi.79.24.15043-15053. By contrast, there is extensive virus in the olfactory bulbs and brain. Brain bleeding corresponds to sites that are high in virus and occurs after peripheral virus is cleared. Furthermore, we present evidence that virus replicates in brain vessels. This issue and reference to these previous publications have been added to the Discussion on page 18, where the possibility of peripheral pathology is discussed.

2. Anti-LFA/VLA-4 blockade used to block peripheral immune cell CNS infiltration revealed that microbleeds were detected earlier and covered larger volumes, which was consistent with impaired viral control. However, the efficacy of the antibody blockade should be confirmed experimentally and noted in results.

We agree. We performed cell counting and flow cytometry experiments to confirm the efficacy of the antibody blockade in this Revision (Figure 3B and Figure 3—figure supplement 1D). The blocking antibodies decreased the number of total immune cells (CD45+) over 95% on day 6.

Comment/Data/Modify:1. Figure 1F – it is stated that the peak in viral loads in the OB is at day 6 however no earlier time points are shown. It would be informative to measure viral loads at day 4 post infection.

Viral titers have been studied from day 1 to day 8 post infection by plaque assay in a previous study [(Moseman et al., 2020) https://pubmed.ncbi.nlm.nih.gov/32503876/, Figure 1E]. We have added this reference and modified our statement in the Revised Manuscript page 6 to 7.

2. Figure 3 – it is important to confirm anti-LFA-1/VLA-4 actually blocked immune cell influx into the brain. Flow cytometry quantitating the absolute number of lymphocytes in the brains with and without treatment would strengthen the results.

We added cell counting and flow cytometry experiments to confirm the efficacy of the antibody blockade in this Revision (Figure 3B and Figure 3—figure supplement 1D). The blocking antibodies decreased the number of total immune cells (CD45+) over 95% on day 6 (Figure 3B). The decreased numbers of neutrophils, Ly6C+, CD8^+^, and CD11b+ cells are shown in Figure 3—figure supplement 1D.

3. Please provide additional quantitative analyses for Figure 4 and 8 on hyperintensities and colocalization with virus or RBC only (Figure 4) and relative frequencies of magnetically labelled T cells present in a pre-vascular compromised region versus in the microbleed, as measured by MRI and histology (Figure 8).

Figure 4 shows that VSV can infect vascular endothelial cells and cause bleeding. Hypointensities were not measured in this Figure. We quantified the numbers of VSV infected vessels, colocalizing and not colocalizing with bleeds. Figure 4D was added with this new data.

Figure 8H was added to show the frequency of MPIO-labeled T cells present with microbleeds versus without microbleeds. In the ONL/GL, 85% of MPIO-labeled T cells were in the region with microbleeds and 15% were in regions with no evidence of bleeding. The MCL/GCL area is where T cells were most often found with no bleeds in the olfactory bulb.

4. Figure 5 – the labelling on this figure is a little confusing (day 6 = early transfer and day 11 late transfer). It would be useful if the authors simplify the labelling to early and late transfer.

Figure 5 labeling was improved in this Revision.

5. Figure 7a – to demonstrate binding specificity of the streptavidin can the authors overlay the histogram of the MPIO- OT-I cells in the culture.

Figure 7B was improved with this data. Figure 7—figure supplement 2A was added to show the FACS method in detail.

6. Figure S9 – The data in this figure is very interesting (T cell dose impacts microbleeding) however the first time Figure S9 brought up is in the discussion – this data needs to be either removed or relocated into the Results section.

These data were relocated into the Results section on page 11.

7. Removing sup Figure 5; The supplementary figures are not presented in chronological order – this should be amended so they are presented in the order they are introduced in the text.

Previous Figure S5 was moved to Figure 5—figure supplement 1 in this Revision.

8. – Lines 81-82: The depiction of "early stage" may not be fitting is this context since T2* changes for microbleeds will persist and are not stageable.– Line 127 indicates microbleeds were already evident in the midbrain beginning at d4 and line 130 again notes microbleeds were observed in the midbrain at days 8-11. The significance of these findings should be stated more clearly.– Line 167, Figure 3H depicts viral RNA , not virus titers.

These corrections and improvements were made in this Revision.

9. Though this is a different antibody, magnetic labeling of T cells by antibody linked MPIOs has been accomplished by Medford et al., and should be acknowledged in the manuscript: https://pubmed.ncbi.nlm.nih.gov/17541955/.

It is true that our method is an extension of the previous work as shown in this reference. This important reference was added on page 12.

10. Figure 6c – validation of this data using a more robust and quantitative read out (flow cytometry of brain tissue measuring OT-I cells numbers) would be useful. Also it is unclear why anti-CD8 was used instead of dTomato to identify the transferred OT-I cells. It would be interesting to distinguish the localisation of the endogenous CD8 and activated OT-I cells in these sections.

Flow cytometric analyses of the olfactory bulbs measuring OT-I cell numbers had been performed and published previously [(Moseman et al., 2020), Figure 2 E-F]. As in this reference, with no OT-I T cell transfer, there are about 2 x 10^5^ (endogenous) CD8^+^ T cells in the OB on day 6. With OT-I T cell transfer, there are about 10^6^ OT-I cells in the OB. In the present manuscript, the IHC shown in Figure 6 is consistent with previous flow cytometry data. The purpose of the IHC study in Figure 6 is to provide more information of the distribution pattern of the virus, bleeds, and CD8 T cells.

We had to use anti-CD8 because mTomato on OT-I cells is not very bright, especially after fixation, as shown in the new Figure 7—figure supplement 2B and C. Moreover, the olfactory bulb has high auto-fluorescence background. The high auto-fluorescence background in the bulb makes the mTomato on OT-I cells even more difficult to detect. Thus, anti-CD8 was used to stain total CD8 T cells.

11. Please provide a control experiment using transfer of labelled CD8 T cells with heterologous specificity (e.g. LCMV, EBV specific) to assess if antigen specificity is required for CD8 T cell signal? This approach also address concerns regarding active migration vs passive accumulation of cells.

Antigen specificity is not required for CD8 T cell entry into the infected CNS. A previous study demonstrated, for example, that LCMV specific P14 CD8 T cells were able to migrate into the VSV infected brain [(Moseman et al., 2020) Figure 2 E-F, Figure 4 B, Figure 5 B].

You should include a reference for 1 pg detection limit for iron per voxel by mri.Is there a reference for the ability to detect 10 RBCs by MRI?

These numbers are calculated from the literature. MRI has the potential to detect a single MPIO particle (1.1 pg iron per particle) (Shapiro et al., 2004) https://doi.org/10.1073/pnas.0403918101. The iron content in 1 RBC was calculated from the reference (Ozment and Turi, 2009) https://pubmed.ncbi.nlm.nih.gov/18992790/. 1 RBC contains ~0.2 pg iron. Thus, MRI shows the potential to detect about 10 RBCs. References were added on page 17 of the Revision.

The magnetic susceptibility generated by deoxygenated blood is orders of magnitude below that of MPIOs. Is there no way reduce the TE in the MRI experiment to substantially reduce the signal from the microbleeds and keep the signal of the magnetically labeled cells? Or to additionally perform T2-weighted imaging to minimize the signal from the MPIOs?

Based on the last response we think it takes only 5 to 10 red blood cells to equal an MPIO. Thus, a little bleeding is hard to distinguish from a few T cells even with a quantitative T2* approach. However, this is a critical next step for us since bleeding so often occurs with immune cell infiltration in the brain. We have discussed potential approaches and have added the idea that quantitative T2* analysis especially at different magnetic field strengths may be an approach to accomplish this. It will be crucial for MRI cell tracking under the condition of bleeding, which is one common pathology associated with many diseases.

These points have been suggested. It would be great if data are available but the editorial group understand that they may be beyond the scope of the submission. Please however provide a comment or justification in the revised version and response to reviewers:9. Figure 4 – it is suggested based on the IHC data that VSV infects vascular endothelial cells. To strengthen this claim, I recommend the authors purify out CD31+ endothelial cells and measure VSV RNA in these cells. [may not be achievable]

We hope that the histology result (Figure 4) provided enough evidence to suggest that VSV infects brain vascular endothelial cells. It is challenging to measure VSV RNA in purified CD31+ endothelial cell especially at the early stages of infection. There likely is low levels of virus in CD31+ cells and we do not have quantitative information on how many vessels are infected. At peak stages, virus likely lyses the infected CD31+ cells possibly causing bleeding. It is also likely that virus in the vascular endothelial cells can be detected and cleared to some extent by the peripheral immune system. In future studies we will tackle this challenging problem and determine if endothelial cells have VSV RNA.

Reviewer #1 (Recommendations for the authors):Though this is a different antibody, magnetic labeling of T cells by antibody linked MPIOs has been accomplished by Medford et al., and should be acknowledged in the manuscript: https://pubmed.ncbi.nlm.nih.gov/17541955/.

It is true that our method is an extension of the previous work as shown in this reference. This important reference was added on page 12.

You should include a reference for 1 pg detection limit for iron per voxel by mri.Is there a reference for the ability to detect 10 RBCs by MRI?

These numbers are calculated from the literatures. MRI has the potential to detect single MPIO particle (1.1 pg iron per particle) (Shapiro et al., 2004) https://doi.org/10.1073/pnas.0403918101. The iron content in 1 RBC was calculated from the reference (Ozment and Turi, 2009) https://pubmed.ncbi.nlm.nih.gov/18992790/. 1 RBC contains ~0.2 pg iron. Thus, MRI shows the potential to detect about 10 RBCs. References were added on page 17.

The magnetic susceptibility generated by deoxygenated blood is orders of magnitude below that of MPIOs. Is there no way reduce the TE in the MRI experiment to substantially reduce the signal from the microbleeds and keep the signal of the magnetically labeled cells? Or to additionally perform T2-weighted imaging to minimize the signal from the MPIOs?

Based on the last response we think it takes only 5 to 10 red blood cells to equal an MPIO. This a little bleeding is hard to distinguish from a few T cells even with a quantitative T2* approach. However, this is a critical next step since bleeding so often occurs with immune cell infiltration in the brain. We have discussed potential approaches and have added the idea that quantitative T2* analysis especially at different magnetic field strengths may be an approach to accomplish this. It will be crucial for MRI cell tracking under the condition of bleeding, which is one common pathology associated with many diseases.

Reviewer #2 (Recommendations for the authors):1. More rigorous investigation of potential cerebrovascular microbleeds following systemic or respiratory infections not affecting the CNS would provide a more definitive evidence that direct CNS infection, rather than peripheral pathology, is the causation of microbleeds.

We agree with this reviewer that peripheral pathology could cause CNS bleeds, which is very important. Previous study showed that following intranasal inoculation, VSV replicates in the mouse lung, causes a transient viremia, and is found transiently in other organs, including liver and spleen. The virus is cleared from the periphery within 2-3 days. (Ramsburg et al., 2005; Roberts et al., 1999). Viral titer (Moseman et al., 2020) and histology (Figures 1,2,8) showed that VSV infected brain very efficiently, peaking on day 6 before being cleared on day 8. Microbleeds largely co-localized with sites of VSV replication in the brain. Based on these, we think that direct CNS infection, rather than peripheral pathology, causes brain microbleeds. We added discussion of this point and references on page 18 in this Revision.

2. Experimental: Anti-LFA/VLA-4 blockade used to block peripheral immune cell CNS infiltration revealed that microbleeds were detected earlier and covered larger volumes, which was consistent with impaired viral control. However, the efficacy of the antibody blockade should be confirmed experimentally and noted in results.

We agree. We added cell counting and flow cytometry experiments to confirm the efficacy of the antibody blockade in this Revision (Figure 3B and Figure 3—figure supplement 1D). The blocking antibodies decreased the number of total immune cells (CD45+) over 95% on day 6 (Figure 3B). The decreased numbers of neutrophils, Ly6C+, CD8^+^, and CD11b+ cells are shown in Figure 3—figure supplement 1D.

Reviewer #3 (Recommendations for the authors):1. This is an interesting paper, and the capacity to track virus specific T cells and cerebrovascular pathology in real time in living brains would be a major technological advance. However, the model proposed by the authors does not achieve this. It was necessary in this system to perform IHC on fixed brain sections to differentiate T cell influx and microbleeds

We thank this Reviewer for agreeing with us that “track virus specific T cells and cerebrovascular pathology in real time in living brains would be a major technological advance”. It is our goal by using this well studied model. We were very surprised to see that cerebrovascular breakdown and microbleeds take place even at day 1 of infection through MRI studies. This new finding poses challenges for the goal for tracking T cells. However, it shows the importance of MRI guided histology in the study of cerebrovascular pathology and neuroinflammation. We have added new data since the original review (new Figure 8I) that show time lapse imaging of the deep OB region prior to bleeding from 20-24 hour. We can detect the earliest arrival of T cells to this region. There was no evidence of migration from the surface ONL/GL regions of the bulb, and once in the deep OCB regions we saw no significant migration on the distance scale of MRI. This leads to the model that T cells are entering via local extravasation through the vasculature. T cell entry, vascular breakdown and hemorrhage are common features. We think this work takes a significant step forward to describing the sequence of event in this model of viral infection that will be useful for many other modes. Future MRI developments to distinguish bleeds from MPIOs will be crucial, and we have discussed where we think those developments will come from in the near future.

2. Figure 1F – it is stated that the peak in viral loads in the OB is at day 6 however no earlier time points are shown. It would be informative to measure viral loads at day 4 post infection.

Viral titers have been studied from day 1 to day 8 post infection by plaque assay in previous study [(Moseman et al., 2020) https://pubmed.ncbi.nlm.nih.gov/32503876/, Figure 1E]. We have added this reference and modified our statement in the Revised Manuscript page 6 to 7.

3. Figure 3 – it is important to confirm anti-LFA-1/VLA-4 actually blocked immune cell influx into the brain. Flow cytometry quantitating the absolute number of lymphocytes in the brains with and without treatment would strengthen the results.

We added cell counting and flow cytometry experiments to confirm the efficacy of the antibody blockade in this Revision (Figure 3B and Figure 3—figure supplement 1D). The blocking antibodies decreased the number of total immune cells (CD45+) over 95% on day 6 (Figure 3B). The decreased numbers of neutrophils, Ly6C+, CD8^+^, and CD11b+ cells are shown in Figure 3—figure supplement 1D.

4. Figure 4 – it is suggested based on the IHC data that VSV infects vascular endothelial cells. To strengthen this claim, I recommend the authors purify out CD31+ endothelial cells and measure VSV RNA in these cells.

We hope that the histology result (Figure 4) provided enough evidence to suggest that VSV infects brain vascular endothelial cells. It is challenging to measure VSV RNA in purified CD31+ endothelial cell especially at the early stages of infection. There likely is low levels of virus in CD31+ cells, and we do not have quantitative information on how many vessels are infected. At peak stages, virus likely lyses the infected CD31+ cells possibly causing bleeding. It is also likely that virus in the vascular endothelial cells can be detected and cleared to some extent by the peripheral immune system. In future studies we will tackle this challenging problem and determine if endothelial cells have VSV RNA.

5. Figure 5 – the labelling on this figure is a little confusing (day 6 = early transfer and day 11 late transfer). It would be useful if the authors simplify the labelling to early and late transfer.

Figure 5 was improved in the Revision.

6. Figure 6c – validation of this data using a more robust and quantitative read out (flow cytometry of brain tissue measuring OT-I cells numbers) would be useful. Also it is unclear why anti-CD8 was used instead of dTomato to identify the transferred OT-I cells. It would be interesting to distinguish the localisation of the endogenous CD8 and activated OT-I cells in these sections.

Flow cytometry analysis of olfactory bulb measuring OT-I cell numbers has been performed and published previously [(Moseman et al., 2020), Figure 2 E-F]. As in this reference, with no OT-I T cell transfer, there are about 2 x 10^5^ (endogenous) CD8^+^ T cells in the OB on day 6. With OT-I T cell transfer, there are about 10^6^ OT-I cells in the OB. In this manuscript, the IHC showing in Figure 6 is consistent with previous flow cytometry data. The purpose of the IHC study in Figure 6 is to provide more information of the distribution pattern of the virus, bleeds, and CD8 T cells.

We have to use anti-CD8 because mTomato on OT-I cells is not very bright, especially after fixation, as shown in the new Figure 7—figure supplement 2B and C. Moreover, olfactory bulb shows high auto-fluorescence background. The high auto-fluorescence background in the bulb makes the mTomato on OT-I cells even more difficult to detect. Thus, anti-CD8 was used to stain total CD8 T cells.

7. Figure 7a – to demonstrate binding specificity of the streptavidin can the authors overlay the histogram of the MPIO- OT-I cells in the culture.

Figure 7B was improved with this data. Figure 7—figure supplement 2A was added to show the FACS method in detail.

8. Figure S9 – The data in this figure is very interesting (T cell dose impacts microbleeding) however the first time Figure S9 brought up is in the discussion – this data needs to be either removed or relocated into the Results section.

This data was relocated into the Results section on page 11.

References:

Herz, J., Johnson, K. R., and McGavern, D. B. (2015). Therapeutic antiviral T cells noncytopathically clear persistently infected microglia after conversion into antigen-presenting cells. Journal of Experimental Medicine, 212(8), 1153-1169. https://doi.org/10.1084/jem.20142047

Moseman, E. A., Blanchard, A. C., Nayak, D., and McGavern, D. B. (2020). T cell engagement of cross-presenting microglia protects the brain from a nasal virus infection. Science Immunology, 5(48). https://doi.org/ARTN eabb181710.1126/sciimmunol.abb1817

Ozment, C. P., and Turi, J. L. (2009). Iron overload following red blood cell transfusion and its impact on disease severity. Biochim Biophys Acta, 1790(7), 694-701. https://doi.org/10.1016/j.bbagen.2008.09.010

Ramsburg, E., Publicover, J., Buonocore, L., Poholek, A., Robek, M., Palin, A., and Rose, J. K. (2005). A vesicular stomatitis virus recombinant expressing granulocyte-macrophage colony-stimulating factor induces enhanced T-cell responses and is highly attenuated for replication in animals. Journal of Virology, 79(24), 15043-15053. https://doi.org/10.1128/Jvi.79.24.15043-15053.2005

Roberts, A., Buonocore, L., Price, R., Forman, J., and Rose, J. K. (1999). Attenuated vesicular stomatitis viruses as vaccine vectors. Journal of Virology, 73(5), 3723-3732. https://doi.org/Doi 10.1128/Jvi.73.5.3723-3732.1999

Shapiro, E. M., Skrtic, S., Sharer, K., Hill, J. M., Dunbar, C. E., and Koretsky, A. P. (2004). MRI detection of single particles for cellular imaging. Proc Natl Acad Sci U S A, 101(30), 10901-10906. https://doi.org/10.1073/pnas.0403918101